# Shifting Time: Time-series Forecasting with Khatri-Rao Neural Operators

**Srinath Dama** [1]   **Kevin Course** [1]   **Prasanth B. Nair** [1]

## Abstract

We present an operator-theoretic framework for temporal and spatio-temporal forecasting based on learning a *continuous time-shift operator*. Our operator learning paradigm offers a continuous relaxation of the discrete lag factor used in traditional autoregressive models, enabling the history of a system up to a given time to be mapped to its future values. We parametrize the time-shift operator using Khatri-Rao neural operators (KRNOs), a novel architecture based on non-stationary integral transforms with nearly linear computational scaling. Our framework naturally handles irregularly sampled observations and enables forecasting at super-resolution in both space and time. Extensive numerical studies across diverse temporal and spatio-temporal benchmarks demonstrate that our approach achieves state-of-the-art or competitive performance with leading methods.

## 1. Introduction

Time series forecasting is a fundamental problem in machine learning and statistics with applications to a broad spectrum of problems encountered in all branches of science, engineering, and finance (Roberts et al., 2013; Milani et al., 2017; Siami-Namini & Namin, 2018). At a high-level, time-series problems are concerned with forecasting the future values of quantities of interest given past observations of the same or correlated quantities.

The majority of methods for time-series forecasting largely fall into the categories of autoregressive moving average models and their variants (Box & Jenkins, 1976; Girard, 2004), and deep autoregressive models with memory (Elman, 1990; Hochreiter & Schmidhuber, 1997; Salinas et al., 2020). With the tremendous success of transformer-based

[1]Institute for Aerospace Studies, University of Toronto, ON, Canada. Correspondence to: Srinath Dama <srinath.dama@mail.utoronto.ca>, Kevin Course <kevin.course@mail.utoronto.ca>, Prasanth B. Nair <prasanth.nair@utoronto.ca>.

*Proceedings of the $42^{nd}$ International Conference on Machine Learning*, Vancouver, Canada. PMLR 267, 2025. Copyright 2025 by the author(s).

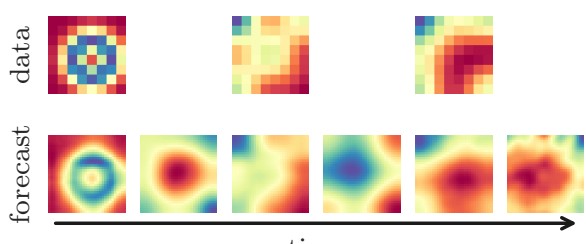

*Figure 1.* The top row shows low-resolution test data. In the bottom row we plot a high-resolution forecast. By parametrizing the time-shift operator with a Khatri-Rao neural operator we can forecast in super-resolution in both *space* and *time*.

models in natural language processing tasks (Vaswani et al., 2017) and computer vision applications (Dosovitskiy et al., 2020), this class of models is gaining popularity in time-series forecasting (Chen et al., 2021; Nie et al., 2023; Zhou et al., 2022; Wu et al., 2023; Liu et al., 2022; 2024; Gruver et al., 2024; Zhang et al., 2024). In the world of spatio-temporal forecasting, Gaussian processes (Hamelijnck et al., 2021), deep operator networks (DeepONets) (Lu et al., 2021), and neural operators (Li et al., 2020b;c;d) have emerged as cornerstones of the literature.

A major challenge with all autoregressive-style models is that observations are required to be provided at a constant frequency at both training and inference time. This requirement introduces a number of challenges in practice. First, when observations are not provided at regular intervals, it is common practice to create a hierarchy of approximations that can negatively impact performance for reasons unrelated to the capacity of the model. Second, in an online setting, this requirement necessitates creating a pipeline for imputing any missing data points (due to sensor error or system latency) before predictions can be made. While neural ordinary differential equations based methods (Chen et al., 2018; Rubanova et al., 2019) have shown tremendous promise for learning from irregularly spaced observations, they are challenging to scale and train for large-scale temporal and spatio-temporal datasets.

In the present work, we propose casting time-series forecasting problems as a supervised learning problem of the *continuous time-shift operator*. In contrast to standard autoregressive models based on discrete-time (or discrete space-time) representation of the dynamics, the continuous time-shift

operator maps the entire, continuous history of the dynamics over a past time-window into its future values over a subsequent time-window. Our operator-theoretic approach can be viewed as a continuous relaxation of the discrete lag factor in autoregressive models. This offers several practical advantages such as the ability to learn directly from irregularly sampled observations and to forecast at super-resolution in both space and time while retaining the stability of training neural operators; see Figure 1.

To address the complexities of learning the time-shift operator for temporal and spatio-temporal dynamical systems, we propose Khatri-Rao neural operators (KRNOs). KRNOs are a new architecture for operator learning based on non-stationary integral transforms that provides exceptional model flexibility compared to methods based on stationary kernels (Li et al., 2020b; Kovachki et al., 2023; Rahman et al., 2023), while achieving almost linear scaling. We demonstrate the efficacy of the proposed approach on a suite of challenging test cases, including shallow water simulation (Kissas et al., 2022), a climate modeling problem (Kissas et al., 2022), a set of challenging irregularly sampled time-series benchmarks, the Darts datasets (Herzen et al., 2022), and the M4 dataset (Makridakis et al., 2020). In total, we evaluate KRNOs on 39 different test cases and compare performance against numerous modern approaches for temporal and spatio-temporal forecasting to demonstrate its strong generalization capabilities.

## 2. Method

We first introduce the continuous time-shift operator for temporal and spatio-temporal dynamical systems. Following this, we propose Khatri-Rao neural operators for learning the time-shift operator.

### 2.1. The continuous time-shift operator

Consider an ordinary differential equation (ODE) $\dot{z}(t) = F(z(t)), z(0) = z_0$, with Lipschitz continuous $F : \mathbb{R}^n \to \mathbb{R}^n$ on the time interval $[0, T]$. While the classical flow map propagates a single state to future times, learning such a map can be challenging in real-world applications involving sparse, noisy measurements. We therefore propose an alternative paradigm: the *continuous time-shift operator* $\mathcal{A}_{t_p}^{t,t_f}$, a causal, continuous-time operator that maps the history of $z$ over $[t_p, t]$ to its future values over $(t, t_f]$, where $0 \leq t_p < t < t_f \leq T$, i.e., a propagator of the form

$$z(\tau) = (\mathcal{A}_{t_p}^{t,t_f} z)(\tau), \ \forall \tau \in (t, t_f]. \qquad (1)$$

The existence of $\mathcal{A}_{t_p}^{t,t_f} : L^2([t_p, t]; \mathbb{R}^n) \to L^2((t, t_f]; \mathbb{R}^n)$ follows from the Picard-Lindelöf theorem and noting that $z(\tau) = z(t) + \int_t^\tau F(z(s))ds, \ \tau \in (t, t_f]$. The time-shift operator satisfies two key properties: (1) semigroup

property: $\mathcal{A}_{t_p}^{t_2,t_f} = \mathcal{A}_{t_1}^{t_2,t_f} \circ \mathcal{A}_{t_p}^{t_1,t_2}$, where $t_p < t_1 < t_2 < t_f$, and (2) continuity property: $\exists C > 0$ such that $||\mathcal{A}_{t_p}^{t,t_f} z_1 - \mathcal{A}_{t_p}^{t,t_f} z_2||_{L^2((t,t_f];\mathbb{R}^n)} \leq C||z_1 - z_2||_{L^2([t_p,t];\mathbb{R}^n)}$ for all $z_1, z_2 \in L^2([t_p, t]; \mathbb{R}^n)$; see Appendix A for a proof.

The proposed time-shift operator formalism offers several compelling advantages for data-driven learning. First, it offers robustness through leveraging a richer representation of the system's history, potentially mitigating uncertainty in any single observation. Second, it naturally accommodates irregularly sampled observations by treating the history of system dynamics as a function. In other words, since the time-shift operator is continuous in time, it can be learned from irregularly sampled observations (similar to neural ODEs (Chen et al., 2018) but without requiring adjoint sensitivity calculations), a significant advantage in many practical applications. Third, the continuous-time formulation enables super-resolution forecasting in both space and time. Moreover, since the operator depends on $t_p$ and $t_f$, this representation enables the study of the dynamics of complex systems over different time scales. In time-series forecasting contexts, $t_p$ and $t_f$ are treated as hyperparameters that can be tuned via cross-validation or using techniques such as hypergradient descent (Chandra et al., 2022).

The notion of shift operators has been widely studied in functional analysis; see, for example, Marchenko (2006). Recent theoretical work (Zhen et al., 2022b;a) leveraged time-shift operators while studying the relationship between the spectra of the autocorrelation function and the infinite-dimensional Koopman operator (Koopman, 1931b) governing the evolution of observables. However, to the best of our knowledge, the idea of developing an operator-theoretic framework to directly learn the continuous time-shift operator from time-series and spatio-temporal observations remains unexplored.

In the present work, we propose to parametrize the time-shift operator using a neural operator. To motivate this, consider the special case where $\mathcal{A}_{t_p}^{t,t_f} : L^2([t_p, t]; \mathbb{R}^n) \to L^2((t, t_f]; \mathbb{R}^n)$ is a Hilbert-Schmidt operator (Retherford, 1993). Then there exists a kernel $\kappa : [0, T] \times [0, T] \to \mathbb{R}$ satisfying the condition

$$z(\tau) = (\mathcal{A}_{t_p}^{t,t_f} z)(\tau) = \int_{t_p}^t \kappa(\tau, s) z(s) ds, \qquad (2)$$

$\forall \tau \in (t, t_f]$, where the dependence of the kernel on $(t, t_p, t_f)$ is not explicitly indicated for simplicity of notation. It is worth noting that even though the preceding continuous convolution integral representation holds under restrictive assumptions on the dynamics, this representation motivates approximating the time-shift operator for general nonlinear dynamical systems using deep neural operators, which involve a nested composition of integral transforms and point-wise nonlinearities.

We can similarly define the spatio-temporal time-shift operator for a scalar field $u : \Omega \times [0, T] \to \mathbb{R}$, where $\Omega \subset \mathbb{R}^{d-1}$ ($d > 1$) denotes a bounded Lipschitz domain. Using the non-overlapping time-intervals defined previously, the spatio-temporal time-shift operator can be defined as

$$u(x, \tau) = (\mathcal{A}_{t_p}^{t, t_f} u)(x, \tau), \quad \forall x \in \Omega, \ \tau \in (t, t_f]. \quad (3)$$

Under the assumption that $u$ lies in the separable Hilbert space $\mathcal{U}(\Omega \times [0, T]; \mathbb{R})$, and assuming the spatio-temporal time-shift operator is a Hilbert-Schmidt operator mapping from $\mathcal{U}(\Omega \times [t_p, t]; \mathbb{R})$ to $\mathcal{U}(\Omega \times (t, t_f]; \mathbb{R})$, we have the following integral representation

$$u(x, \tau) = (\mathcal{A}_{t_p}^{t, t_f} u)(x, \tau), \ x \in \Omega, \ \tau \in (t, t_f]$$
$$= \int_{\Omega} \int_{t_p}^{t} \kappa(\{x, \tau\}, \{y, s\}) u(y, s) dy ds, \quad (4)$$

where $\kappa : \Omega \times [0, T] \times \Omega \times [0, T] \to \mathbb{R}$ is a square-integrable kernel. The preceding representation in terms of an integral transform motivates the application of deep neural operators to approximate the time-shift operator of complex spatio-temporal dynamical systems. Furthermore, universal approximation results for neural operators (Kovachki et al., 2023) guarantee that, under suitable regularity assumptions, the time-shift operator can be well approximated.

It is worth noting that Li et al. (2020b) considered an operator learning test problem where a two-dimensional flow field over the time-interval $[0, 10]$ is mapped to the interval $(10, T]$ (for a fixed $T$). This was tackled using an autoregressive Fourier neural operator (FNO) model and a 3D FNO model, with the latter model making predictions over the entire spatio-temporal domain of interest. The time-shift operator learning formalism presented here allows us to view the test-case involving FNO-3D in (Li et al., 2020b) as a special case of the general setting considered here with a stationary-kernel based neural operator parametrization of the time-shift operator and fixed values of $(t_p, t_f)$. In the next section, we present a new architecture for parametrizing the time-shift operator that enables over an order of magnitude reduction in the number of parameters compared to FNO, while achieving superior accuracy.

## 2.2. Khatri-Rao neural operators (KRNOs)

We now introduce KRNOs, a new operator learning architecture based on non-stationary integral transforms, to approximate the time-shift operator of temporal and spatio-temporal dynamical systems. KRNOs offer expressive parametrization of operators using non-stationary integral transform layers which (i) do not require any approximation of the kernel and (ii) scale almost linearly in the number of quadrature nodes. As far as we are aware, ours is the only approach for parametrizing neural operators which combines

*Table 1.* Comparison of Khatri-Rao Neural Operator (KRNO), Graph Neural Operator (GNO) (Li et al., 2020c), Multipole Graph Neural Operator (MGNO) (Li et al., 2020d), and Fourier Neural Operator (FNO) (Li et al., 2020b), for computing kernel integral transforms, as compiled by (Kovachki et al., 2023). Here $N' << N$ is a constant used to control the variance of the integral transform approximation. Ours is the only approach which allows for exact, non-stationary kernel evaluations while achieving almost linear computational cost.

| Method | Time | Non-stationary | Exact kernel[1] |
|--------|------|:--------------:|:---------------:|
| GNO | $\mathcal{O}(NN')$ | ✓ | ✗ |
| MGNO | $\mathcal{O}(N)$ | ✓ | ✗ |
| FNO | $\mathcal{O}(N \log N)$ | ✗ | ✓ |
| KRNO | $\mathcal{O}(N^{1+1/d})$ | ✓ | ✓ |

these advantages. We will show later that KRNOs provide state-of-the-art performance across a number of benchmarks while inheriting the benefits of neural operators such as being discretization-invariant and enabling super-resolution in forecasts (Li et al., 2020b).

**Neural operators** Neural operators (NOs) are an expressive class of models for approximating maps between function spaces. In contrast to standard multi-layer perceptrons, which are defined by an alternating series of affine maps and nonlinear activations, NOs are defined by an alternating series of linear, kernel integral transforms and nonlinear activations. For simplicity of exposition, consider an integral transform layer (Li et al., 2020b; Kovachki et al., 2023) that maps the input spatio-temporal vector field $v_\ell : \Omega \times [0, \tau] \to \mathbb{R}^p$ to $v_{\ell+1} : \Omega \times [0, \tau] \to \mathbb{R}^q$, defined below

$$v_{\ell+1}(t, x) = \mathcal{K}(v_\ell)(t, x)$$
$$= \int_{\Omega} \int_0^{\tau} \kappa(\{t, x\}, \{t', x'\}) v_\ell(t', x') dt' dx'$$
$$+ W v_\ell(t, x) + b, \quad (5)$$

where $\kappa : \mathbb{R} \times \Omega \times \mathbb{R} \times \Omega \to \mathbb{R}^{q \times p}$ is a matrix-valued kernel, $W \in \mathbb{R}^{q \times p}$ is a weight matrix, and $b \in \mathbb{R}^q$ is a bias vector. It is also common to prepend and append the preceding layer by a series of point-wise lifting and projection layers (Kovachki et al., 2023). Note that in (5), the inputs and outputs are assumed to be defined over the same spatio-temporal domain for simplicity – we will later consider the general case when the input and output domains are different.

Rather than computing the integral transforms exactly, NOs propagate evaluations of the intermediate functions at a set of quadrature nodes through the network. Let $X = \{\{t_1, x_1\}, \{t_2, x_2\}, \ldots, \{t_N, x_N\}\} \in \mathbb{R}^{N \times d}$, where

---

[1]We use the term "Exact Kernel" to indicate that the only source of approximation error stems from the quadrature scheme; the non-stationary kernel itself is evaluated without further approximation.

$t_i \in \mathbb{R}$ and $x_i \in \mathbb{R}^{d-1}$, $i = 1, \ldots, N$ denote quadrature nodes in the $d$-dimensional spatio-temporal domain ($N = n^d$, where $n$ is the number of quadrature nodes per dimension), and let $w \in \mathbb{R}^N$ denote the vector of quadrature weights. As we will show, while computing the point-wise transformation defined by the weights $W$ and $b$ scales as $\mathcal{O}(qpN)$, the primary computational bottleneck in computing the output from a kernel integral transform layer arises from approximating the integral over the domain of the input function.

In the discussion on computational complexity that follows we omit writing the dependence on $q$ and $p$ since these are architectural considerations and the required value for $N$ will be dependent on the complexity of the input function. Letting $v_\ell(X) \in \mathbb{R}^{N \times p}$ be the $\ell^{\text{th}}$ layer evaluated at the quadrature nodes, the kernel integral transform can be approximated as

$$\int_\Omega \int_0^\tau \kappa(X, \{t', x'\}) v_\ell(t', x') dt' dx'$$
$$\approx \kappa(X, X) \text{vec}(\text{diag}(w) v_\ell(X)), \quad (6)$$

where vec : $\mathbb{R}^{N \times p} \to \mathbb{R}^{Np}$ creates a vector from a matrix by stacking columns, diag : $\mathbb{R}^N \to \mathbb{R}^{N \times N}$ converts a vector into a diagonal matrix, $\kappa(X, \{t', x'\}) \in \mathbb{R}^{Nq \times p}$ represents the kernel evaluated between all the quadrature nodes $X$ in the output domain and a single node $\{t', x'\}$ in the input domain. Meanwhile, $\kappa(X, X) \in \mathbb{R}^{Nq \times Np}$ is the kernel evaluated between all the quadrature nodes $X$ in both the output and input domains.

Naively evaluating this kernel integral transform scales as $\mathcal{O}(N^2)$ which is prohibitively expensive for even a modest number of quadrature nodes. In light of this computational challenge, a number of approaches have been developed including Graph Neural Operators (Li et al., 2020c), Multipole Graph Neural Operators (Li et al., 2020d), and Fourier Neural Operators (FNOs) (Li et al., 2020b; Rahman et al., 2023). As we will show, our approach is the only one which scales almost linearly in the number of quadrature nodes while enabling non-stationary integral transforms with exact kernel evaluations.

**Khatri-Rao product structure** In order to achieve almost linear scaling in $N$ without having to approximate the kernel function, we assume that the kernel function decomposes as an element-wise product as follows:

$$\kappa(\{t, x\}, \{t', x'\}) = \kappa^{(1)}(t, t') \odot$$
$$\left( \odot_{i=2}^d \kappa^{(i)}([x]_{i-1}, [x]'_{i-1}) \right), \quad (7)$$

where $\kappa^{(i)} : \mathbb{R} \times \mathbb{R} \to \mathbb{R}^{q \times p}$ for $i = 1, \ldots, d$, $\odot$ denotes the element-wise product, and $[x]_i \in \mathbb{R}$ indicates the $i^{\text{th}}$ element of $x$. While this assumption may appear limiting at first glance, it has been applied extensively in the

context of Gaussian process (GP) regression to build new positive definite kernels and to scale GP regression on product grids (Saatçi, 2011; Wilson et al., 2014). For example, the squared exponential kernel, the Matérn class of kernels, and the spectral mixture product kernel all decompose as a product of the form in Equation (7).

**Proposition 1.** *If the quadrature nodes lie on a product grid, $X = \bar{t} \times x^{(1)} \times \ldots x^{(d-1)}$, where $\bar{t} \in \mathbb{R}^n$ and $x^{(i)} \in \mathbb{R}^n$ denote the quadrature nodes along the time dimension and the $i^{\text{th}}$ dimension of the spatial coordinate $x$, respectively, and the kernel function has a component-wise product structure of the form given in Equation (7), then the kernel function evaluated at the quadrature nodes inherits the Khatri-Rao product structure,*

$$\kappa(X, X) = \kappa^{(1)}(\bar{t}, \bar{t}) * \left( \overset{d}{\underset{i=2}{*}} \kappa^{(i)}(x^{(i-1)}, x^{(i-1)}) \right), \quad (8)$$

*where $\kappa^{(i)}(\cdot, \cdot) \in \mathbb{R}^{qn \times pn}$ is a block-partitioned matrix where block $jk$ is the $jk^{\text{th}}$ output from the component kernel $\kappa^{(i)}$ evaluated on the outer product of the quadrature nodes along the $i^{\text{th}}$ dimension.*

A proof for Proposition 1 can be found in Appendix B. A practical consequence of this result is that the computational complexity associated with computing kernel integral transforms can be reduced from $\mathcal{O}(N^2)$ to $\mathcal{O}(N^{1+1/d})$ with $\mathcal{O}(N^{2/d} + N)$ memory; see Appendix D for a detailed discussion. These complexity estimates have a linear dependence on $pq$, which are architectural parameters common to all NO approaches. Table 1 provides a comparison of KRNOs to other NOs in the literature. To reiterate what was mentioned previously, ours is the only approach which achieves almost linear cost while enabling non-stationary integral transforms without having to approximate the kernel function.

In Appendix C, we present a generalization of Proposition 1 showing that the same computational advantage also holds when the input and output domains are different with different resolution quadrature grids. This generalized result enables the design of non-stationary integral transform layers that can be viewed as a continuous analog of upsampling and downsampling techniques commonly used in convolutional neural networks. In addition, this generalization can be leveraged to significantly reduce computational cost and memory requirements in applications with high spatial or temporal resolutions by using lower-resolution quadrature grids within the intermediate kernel integral layers. We provide detailed numerical studies comparing the training and inference costs of KRNO and FNO across different spatial resolutions in Tables 9,10, as well as Figure 7 in Appendix G.

In this work, we parametrize each component-wise kernel, $\kappa^{(i)} : \mathbb{R} \times \mathbb{R} \to \mathbb{R}^{q \times p}$ by a neural network; see Appendix E

*Table 2.* Performance comparison of different NO methods on Darcy-flow and hyper-elastic problems. Results with $(\cdot)^{\dagger}$, $(\cdot)^{\ddagger}$ are from Lu et al. (2022) and Li et al. (2023), respectively.

| Method | $L^2$ relative error | |
| --- | --- | --- |
| | Darcy-flow | Elasticity |
| FNO | $1.19 \pm 0.05\%^{\dagger}$ | $5.08\%^{\ddagger}$ |
| DeepONet | $1.36 \pm 0.12\%^{\dagger}$ | $9.65\%^{\ddagger}$ |
| KRNO (ours) | $\mathbf{0.96 \pm 0.04\%}$ | $\mathbf{4.66 \pm 0.09\%}$ |

for details. To illustrate the efficacy of KRNO, we first applied our method to the two-dimensional Darcy-flow and hyper-elastic benchmark problems from Lu et al. (2022) and Li et al. (2023), respectively. In these two problems, the goal is to learn a mapping between input and output fields over a two-dimensional spatial domain. Figure 11 in Appendix illustrates the predictions from KRNO for these two problems; see Appendix H.5 for additional details. It can be noted from Table 2 that KRNO provides improved performance over FNO (Li et al., 2020b) and DeepONet (Lu et al., 2021) on both problems. We will later benchmark the performance of KRNO on learning the time-shift operator across a variety of challenging datasets to demonstrate that our approach often provides competitive performance when compared with SOTA methods.

## 3. Numerical studies

In addition to the two spatial modeling problems (Darcy-flow and hyper-elasticity), we evaluated the performance of the proposed method on a suite of challenging temporal and spatio-temporal forecasting problems. These datasets included two spatio-temporal datasets, and 21 diverse time-series datasets (5 multivariate irregular time-series datasets, 8 univariate time-series from the Darts collection, 6 datasets corresponding to different seasonalities from the M4 competition, one multivariate time series corresponding to trading prices of 14 cryptocurrencies, and a bivariate time series containing the positions of NBA basketball players). In total, this amounts to 39 test cases or error metrics. Across these cases, we compare against numerous modern approaches for temporal and spatio-temporal forecasting problems.

For all the problems, we first convert the datasets into necessary input and output sequence pairs for learning the time-shift operator; see Appendix F for details. The default KRNO architecture used in all experiments has 3 kernel integral layers with 20 channels each, lifting and projection layers are parametrized by MLPs with one hidden layer containing 128 hidden units, and the kernels in the integral layers are parametrized by MLPs with 3 hidden layers. The component-wise kernel function used in KRNO is parameterized by a neural network with three hidden layers (see Appendix E). Additional results and experiment setup details are provided in Appendix H. Aggregated

performance statistics of the proposed approach across all test cases are presented in Appendix Table 8. The codebase used to generate the results is available at `https://github.com/srinathdama/ShiftingTime`.

### 3.1. Spatio-temporal forecasting problems

For spatio-temporal problems, we consider shallow water simulation (Kissas et al., 2022), and a climate modeling dataset (Kissas et al., 2022). For evaluation, we use the $L^2$ relative error metric, $L^2$ relative error $= \|u(\cdot, t) - \hat{u}(\cdot, t)\|_{L^2(\Omega)}/\|u(\cdot, t)\|_{L^2(\Omega)}$, where $u(\cdot, t)$ and $\hat{u}(\cdot, t)$ are true and predicted spatial fields at time $t$.

**Shallow water example** Here, the objective is to learn a spatio-temporal operator that is capable of predicting three field variables (fluid column height $\rho$, velocity in the $x_1$-direction $u$, and velocity in the $x_2$-direction $v$) over a future prediction window $(t, t_f)$ using historical data from a look-back window $[t_p, t]$, i.e., $\mathcal{U}(\Omega \times [t_p, t], \mathbb{R}^3) \to \mathcal{U}(\Omega \times (t, t_f], \mathbb{R}^3))$, where $\Omega := (0, 1) \times (0, 1)$ denotes the spatial domain. It is worth noting that the spatio-temporal forecasting problem statement considered here is significantly more challenging than the standard benchmark in prior work, which involves mapping the initial condition to the solution at a single future time (Kissas et al., 2022).

The dataset used for this problem is taken from Kissas et al. (2022), which includes simulated data generated on a $32 \times 32$ spatial grid over the time window $(0, 1)$ and collected at every 0.01 seconds. The training and testing datasets each consist of 1000 simulations with different initial conditions. Both the look-back and prediction window period are set to 0.05 seconds. For evaluation on testing data, we use the three field variables from the first 0.05 seconds window to recursively predict their evolution until 0.6 seconds. As a baseline method, we consider FNO-3D model (Kovachki et al., 2023) and attention based neural operator LOCA (Kissas et al., 2022) to approximate the time-shift operator alongside the proposed KRNO method.

Table 3 and Figure 2 compares the relative $L^2$ error (averaged across 1000 test simulations) for the three field variables when training is conducted for 100 epochs. The results indicate that KRNO delivers superior performance relative to FNO-3D and LOCA. In addition, it is worth noting that KRNO only uses 6% of the parameters required by FNO-3D (see Table 3). Predictions from KRNO for a test simulation are shown in Figure 3. Additional numerical results for this test case can be found in Appendix H.6.

**Climate modeling example** In this experiment, we consider the problem of approximating a spatio-temporal time-shift operator that maps the surface air temperature and surface air pressure, i.e., $\mathcal{U}(\Omega \times [t_p, t]; \mathbb{R}^2) \to \mathcal{U}(\Omega \times (t, t_f]; \mathbb{R}^2))$, where $\Omega := [-90, 90] \times [0, 360]$ denotes the spa-

*Table 3.* Comparison of the average relative $L^2$ errors on the shallow water problem for the three field variables.

| Method | #Parameters | $L^2$ relative error | | |
|---|---|---|---|---|
| | | $\rho$ | $u$ | $v$ |
| FNO-3D | $2,462,895$ | 0.00211 | 0.02606 | 0.02637 |
| LOCA | $94,477,220$ | 0.00314 | 0.15221 | 0.14999 |
| KRNO | $146,159$ | **0.00145** | **0.01497** | **0.01459** |

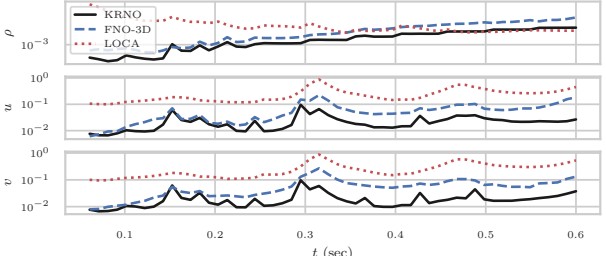

*Figure 2.* Comparison of the average relative $L^2$ errors as a function of time for the three field variables (across the 1000 test simulations) obtained using KRNO, FNO-3D and LOCA models.

tial domain defined in terms of latitude and longitude. The dataset is taken from Kissas et al. (2022) which is based on the Physical Sciences Laboratory meteorological data (Kalnay et al., 1996); see `https://psl.noaa.gov/data/gridded/data. ncep.reanalysis.surface.html`. The training data consists of daily temperature and pressure from 2000 to 2005 (1825 days) over a $72 \times 72$ spatial grid. The test data contains observations from the years 2005 to 2010 on the same grid. The KRNO operator is trained on temperature and pressure data from a 7-day look-back window, with a matching 7-day prediction window. For the evaluation on testing data, we used data from the last week of the previous year and recursively predicted the temperature and pressure fields for the whole year. This is repeated for each year in the testing set. Representative predictions for pressure and temperature are shown in Figure 4 along with the corresponding relative $L^2$ errors. It can be seen that the proposed time-shift operator learning approach performs remarkably well for this dataset.

## 3.2. Temporal forecasting problems

We evaluate the performance of our approach on both regularly and irregularly sampled temporal forecasting problems. For irregular time-series, we use the MuJoCo, MIMIC-III, USHCN, and Human Activity benchmarks, along with a synthetic 2D spiral dataset (details in Appendix H.1). For regularly sampled time-series, we use datasets from the Darts (Herzen et al., 2022) and M4 (Makridakis et al., 2020) collections, in addition to the Crypto (Ticchi et al., 2021)

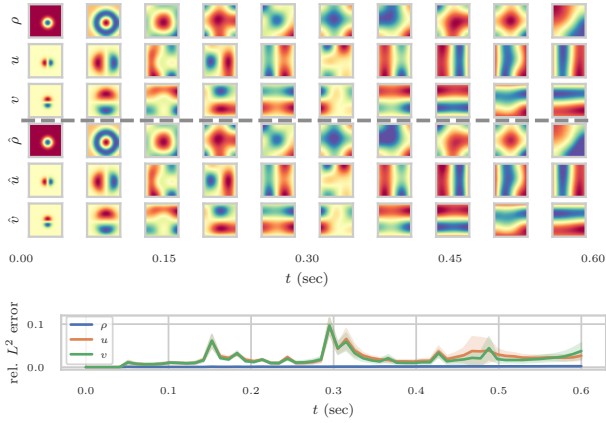

*Figure 3.* Shallow water problem: Top figure shows the predictions $(\hat{\rho}, \hat{u}, \hat{v})$ for the three field variables along with the true fields $(\rho, u, v)$ as a function of time for a test simulation using KRNO trained for 100 epochs. The bottom figure shows the error bars representing the $L^2$ relative errors for three field variables across the 1000 test simulations, with the shaded region indicating $\pm 1$ standard deviation. Additional error plots for the three fields are shown in Figure 13 in the Appendix.

and Player Trajectory datasets.[2]

Distribution shift presents a key challenge in temporal forecasting (Kouw & Loog, 2018; Wang et al., 2021; Kuznetsov & Mohri, 2020). A common practice to tackle this challenge is to use preprocessing strategies, which involve removing known trends and seasonality from the data. To handle distribution shifts in some datasets, we use the reversible instance normalization (ReVIN) approach (Kim et al., 2021) to normalize each input sequence and denormalize the model output predictions.

### 3.2.1. IRREGULARLY SAMPLED TIME-SERIES

We use datasets from diverse domains, including healthcare (MIMIC), climate science (USHCN), biomechanics (Human Activity), and physical simulation (MuJoCo); see Appendix H.0.1 for details. Notably, the MIMIC, USHCN and Human Activity datasets exhibit significant temporal irregularities along with missing values for the states. To handle missing data, we augment KRNOs with one-dimensional convolutional network (CNN) feature extractor module, whose parameters are jointly learned; additional details on the KRNO architecture used are provided in Appendix H.0.1 for details.

Table 4 compares the performance of KRNO with latest SOTA methods such as T-PATCHGNN (Zhang et al., 2024), CRU (Schirmer et al., 2022), Neural Flow (Biloš et al., 2021), and Latent-ODEs (Rubanova et al., 2019). The results highlight the strong performance of KRNO. Specifically, KRNO achieves the best MSE on both the MIMIC

---

[2]`https://github.com/linouk23/NBA-Player-Movements`

*Table 4.* Comparison of KRNO performance with baseline models on irregular time-series datasets. Test MSE and MAE (mean ± std) are shown for each dataset. Top three best results are highlighted in **bold**, underline, and *italic*, respectively. Results for baseline models are sourced from Zhang et al. (2024), Table 1.

| Algorithm | MIMIC | | USHCN | | Human Activity | |
|---|---|---|---|---|---|---|
| | MSE×$10^{-2}$ | MAE×$10^{-2}$ | MSE×$10^{-1}$ | MAE×$10^{-1}$ | MSE×$10^{-3}$ | MAE×$10^{-2}$ |
| DLinear | 4.90 ± 0.00 | 16.29 ± 0.05 | 6.21 ± 0.00 | 3.88 ± 0.02 | 4.03 ± 0.01 | 4.21 ± 0.01 |
| TimesNet | 5.88 ± 0.08 | 13.62 ± 0.07 | 5.58 ± 0.05 | 3.60 ± 0.04 | 3.12 ± 0.01 | 3.56 ± 0.02 |
| PatchTST | 3.78 ± 0.14 | 12.43 ± 0.10 | 5.75 ± 0.01 | 3.57 ± 0.02 | 4.29 ± 0.14 | 4.80 ± 0.09 |
| Crossformer | 2.65 ± 0.10 | 9.56 ± 0.29 | *5.25 ± 0.04* | 3.27 ± 0.09 | 4.29 ± 0.20 | 4.89 ± 0.17 |
| Graph Wavenet | 2.93 ± 0.09 | 10.50 ± 0.15 | 5.29 ± 0.04 | *3.16 ± 0.09* | 2.89 ± 0.03 | *3.40 ± 0.05* |
| MTCNN | 2.71 ± 0.23 | 9.55 ± 0.65 | 5.39 ± 0.05 | 3.34 ± 0.02 | 3.03 ± 0.03 | 3.53 ± 0.03 |
| StemGNN | *1.73 ± 0.02* | 7.71 ± 0.11 | 5.75 ± 0.09 | 3.40 ± 0.09 | 8.81 ± 0.37 | 6.90 ± 0.02 |
| CrossGNN | 2.95 ± 0.16 | 10.82 ± 0.21 | 5.66 ± 0.04 | 3.53 ± 0.05 | 3.03 ± 0.10 | 3.48 ± 0.08 |
| FOURIER/GNN | 2.55 ± 0.03 | 10.22 ± 0.08 | 5.82 ± 0.06 | 3.62 ± 0.07 | 2.99 ± 0.02 | 3.42 ± 0.02 |
| GRU-D | 1.76 ± 0.03 | *7.53 ± 0.09* | 5.54 ± 0.38 | 3.40 ± 0.28 | 2.94 ± 0.05 | 3.51 ± 0.06 |
| SeFT | 1.87 ± 0.01 | 7.84 ± 0.08 | 5.80 ± 0.19 | 3.70 ± 0.11 | 12.20 ± 0.17 | 8.43 ± 0.05 |
| RainDrop | 1.99 ± 0.03 | 8.27 ± 0.07 | 5.78 ± 0.22 | 3.67 ± 0.17 | 14.92 ± 0.14 | 9.45 ± 0.05 |
| Warpformer | 1.73 ± 0.04 | 7.58 ± 0.13 | 5.25 ± 0.05 | 3.23 ± 0.05 | 2.79 ± 0.04 | 3.39 ± 0.03 |
| mTAND | 1.85 ± 0.06 | 7.73 ± 0.13 | 5.33 ± 0.05 | 3.26 ± 0.10 | 3.22 ± 0.07 | 3.81 ± 0.07 |
| Latent-ODE | 1.89 ± 0.19 | 8.11 ± 0.52 | 5.62 ± 0.03 | 3.60 ± 0.12 | 3.34 ± 0.11 | 3.94 ± 0.12 |
| CRU | 1.97 ± 0.02 | 7.93 ± 0.19 | 6.09 ± 0.17 | 3.54 ± 0.18 | 6.97 ± 0.78 | 6.30 ± 0.47 |
| Neural Flow | 1.87 ± 0.05 | 8.03 ± 0.19 | 5.35 ± 0.05 | 3.25 ± 0.05 | 4.05 ± 0.13 | 4.36 ± 0.09 |
| T-PatchGNN | 1.69 ± 0.03 | **7.22 ± 0.09** | 5.00 ± 0.04 | 3.08 ± 0.04 | **2.66 ± 0.03** | **3.15 ± 0.02** |
| KRNO | **1.57 ± 0.02** | 7.43 ± 0.06 | **4.95 ± 0.08** | **3.06 ± 0.08** | *2.85 ± 0.03* | 3.46 ± 0.02 |

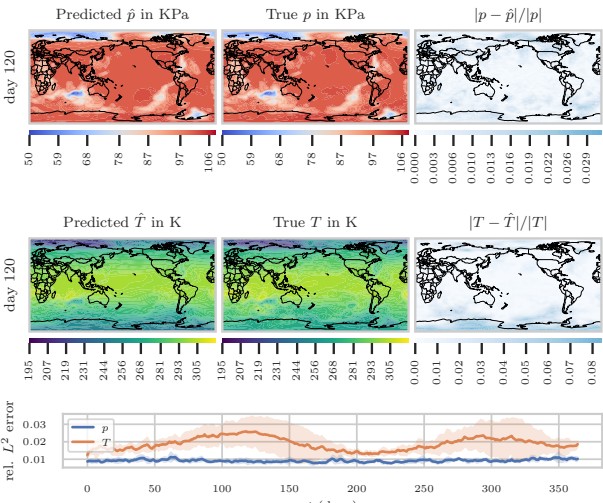

*Figure 4.* Climate modeling problem: Top two figures show the predicted surface pressure and temperature fields using KRNO model along with the true fields for a single day in a forecasted year. Bottom figure shows the error bars representing the $L^2$ relative errors for the five years in test data, with the shaded region indicating ±1 standard deviation.

and USHCN datasets, and also secures the best MAE on USHCN. On the MIMIC dataset, KRNO is the second-best performing model in terms of MAE, closely following T-PatchGNN. For the Human Activity dataset, T-PatchGNN shows the strongest performance, with KRNO ranking third in MSE. Overall, KRNO consistently places among the top-performing methods, demonstrating its robustness and effectiveness for forecasting with complex, irregularly sampled

data. Numerical studies on a synthetic 2D spiral trajectory prediction problem are presented in Appendix H.1.

**MuJoCo** The MuJoCo dataset used in this study is based on simulations of the 'Hopper' model using the DeepMind Control Suite (Tassa et al., 2018). Each trajectory consists of a 14-dimensional state vector sampled at 100 evenly-spaced time points. These trajectories are generated from random initial conditions, allowing the deterministic dynamics of the Hopper model to evolve. We used 100 such trajectories from the dataset provided by Rubanova et al. (2019).

The forecasting objective was to predict the values for the next 10 time steps using observations from the preceding 50 time steps. To simulate irregular sampling, we adopted the preprocessing strategy of Zhang et al. (2024), randomly removing 30%, 50%, or 70% of samples from each trajectory. Table 5 compares KRNO's performance with state-of-the-art (SOTA) methods such as Neural SDEs (Li et al., 2020a; Oh et al., 2024). It can be seen that KRNO outperforms Neural SDEs in all dropout settings. It is worth noting that KRNO achieves an error reduction of 46% on regular data and 38% error reduction on the irregularly sampled versions.

### 3.2.2. REGULARLY SAMPLED TIME-SERIES

**Darts benchmarks** We consider 8 univariate time-series datasets from Darts (Herzen et al., 2022). We compare the performance of the proposed time-shift operator with models such as ARIMA (Box & Jenkins, 1976), neural networks-based models (TCN (Lea et al., 2016), N-BEATS (Oreshkin et al., 2020), N-HiTS (Challu et al., 2023)), SM-GP (Wilson

*Table 5.* Comparison of forecasting performance on the MuJoCo dataset under different percentages of missing observations. Results for baseline methods are sourced from Oh et al. (2024), Table 12.

| Methods | Test MSE | | | |
| --- | --- | --- | --- | --- |
| | Regular | 30% dropped | 50% dropped | 70% dropped |
| GRU-$\Delta t$ | $0.223 \pm 0.020$ | $0.198 \pm 0.036$ | $0.193 \pm 0.015$ | $0.196 \pm 0.028$ |
| GRU-D | $0.578 \pm 0.042$ | $0.608 \pm 0.032$ | $0.587 \pm 0.039$ | $0.579 \pm 0.052$ |
| GRU-ODE | $0.856 \pm 0.016$ | $0.857 \pm 0.015$ | $0.852 \pm 0.015$ | $0.861 \pm 0.015$ |
| ODE-RNN | $0.328 \pm 0.225$ | $0.274 \pm 0.213$ | $0.237 \pm 0.110$ | $0.267 \pm 0.217$ |
| Latent-ODE | $0.029 \pm 0.011$ | $0.056 \pm 0.001$ | $0.055 \pm 0.004$ | $0.058 \pm 0.003$ |
| Augmented-ODE | $0.055 \pm 0.004$ | $0.056 \pm 0.004$ | $0.057 \pm 0.005$ | $0.057 \pm 0.005$ |
| ACE-NODE | $0.039 \pm 0.003$ | $0.053 \pm 0.007$ | $0.053 \pm 0.005$ | $0.052 \pm 0.006$ |
| NCDE | $0.028 \pm 0.002$ | $0.027 \pm 0.000$ | $0.027 \pm 0.001$ | $0.026 \pm 0.001$ |
| ANCDE | $0.026 \pm 0.001$ | $0.025 \pm 0.001$ | $0.025 \pm 0.001$ | $0.024 \pm 0.001$ |
| EXIT | $0.026 \pm 0.000$ | $0.025 \pm 0.004$ | $0.026 \pm 0.000$ | $0.026 \pm 0.001$ |
| LEAP | $0.022 \pm 0.002$ | $0.022 \pm 0.001$ | $0.022 \pm 0.002$ | $0.022 \pm 0.001$ |
| Neural SDE | $0.028 \pm 0.004$ | $0.029 \pm 0.001$ | $0.029 \pm 0.001$ | $0.027 \pm 0.000$ |
| Neural LSDE | $\underline{0.013 \pm 0.000}$ | $0.014 \pm 0.001$ | $0.014 \pm 0.000$ | $\underline{0.013 \pm 0.001}$ |
| Neural LNSDE | $0.012 \pm 0.001$ | $0.014 \pm 0.001$ | $0.014 \pm 0.001$ | $0.014 \pm 0.000$ |
| Neural GSDE | $0.013 \pm 0.001$ | $\underline{0.013 \pm 0.001}$ | $\underline{0.013 \pm 0.000}$ | $0.014 \pm 0.000$ |
| KRNO | $\mathbf{0.007 \pm 0.002}$ | $\mathbf{0.008 \pm 0.002}$ | $\mathbf{0.0114 \pm 0.004}$ | $\mathbf{0.0115 \pm 0.002}$ |

& Adams, 2013) and LLMTIME (Gruver et al., 2024). We used normalized mean absolute error (NMAE)(22) as the evaluation metric. Figure 5 shows the testing errors from KRNO in comparison to other baseline methods presented in (Gruver et al., 2024). The time-shift operator is among the top 3 performing methods on 5 out of 8 datasets in the Darts collection: see Table 8 in Appendix for details.

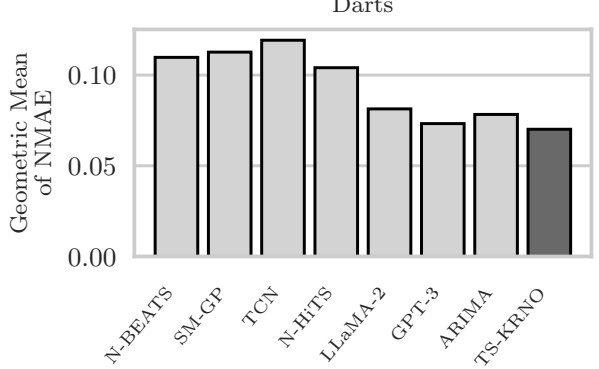

*Figure 5.* Comparison of geometric mean of normalized MAE on univariate Darts datasets for various methods.

**M4 benchmarks** The M4 dataset (Makridakis et al., 2020) is a collection of 100,000 univariate time series from diverse domains such as finance and demographics. This collection comprises six datasets corresponding to different seasonalities, varying from hourly to yearly. On this challenging dataset, the top winning methods in the M4 competition, Smyl (2020) and Montero et al. (2020), Koopman Neural Forecaster (KNF) (Wang et al., 2023), and Nbeats-I+G (Oreshkin et al., 2020) are considered as baselines. All the

*Table 6.* Comparison of sMAPE from KRNO method with other baseline methods for M4. Results with $(\cdot)^\dagger$ were taken from Wang et al. (2023).

| Method | Quarterly | Weekly | Daily |
| --- | --- | --- | --- |
| Montero et al. (2020) | 9.733 | 7.625[†] | 3.097[†] |
| Smyl (2020) | 9.679 | 7.817[†] | 3.170[†] |
| Nbeats-I+G | **9.212** | - | - |
| KNF | 10.008[†] | 7.254[†] | **2.990**[†] |
| KRNO (ours) | 10.503 | **6.934** | 3.086 |

models are evaluated using the symmetric mean absolute percentage error (sMAPE) metric used in the M4 competition. A comparison of KRNO performance on M4 data is presented in Table 6 (for full results see Table 13 in Appendix). We observe that KRNO is among the top two methods on datasets such as M4-Weekly and M4-Daily, where seasonality trends are not present (Wang et al., 2023).

**Crypto and Player Trajectory datasets** The Crypto (Ticchi et al., 2021) dataset is a multivariate time series containing eight features corresponding to trading prices of 14 cryptocurrencies. The objective is to forecast the returns for all 14 cryptocurrencies. The Player Trajectory dataset is a bivariate time series containing the positions of NBA basketball players. The goal here is to forecast the positions of the players. For both datasets, we utilized the same training, validation, and test data as used by Wang et al. (2023). We employed weighted RMSE (Ticchi et al., 2021) for the Crypto data and RMSE for the Player Trajectory data for evaluation. Our method is compared with KNF and other latest methods such as FedFormer (Zhou et al., 2022), Long Expressive Memory (LEM) (Rusch et al., 2022), Variational

Beam Search (VBS) (Li et al., 2021), Multilayer Perceptron (MLP) (Faloutsos et al., 2018), and Vector ARIMA (VARIMA) (Stock & Watson, 2001). Table 7 compares KRNO with these baseline methods. KRNO is the second-best method after KNF on Crypto and Player Trajectory datasets (for full results see Table 15 in Appendix H.4).

*Table 7.* Comparison of RMSE from KRNO method with other baseline methods on Crypto and Player Trajectory datasets.

|  | Crypto (Weighted RMSE $10^{-3}$) | Basketball (RMSE) |
|---|---|---|
| VARIMA | $8.76_{\pm 0.00}$ | $1.26_{\pm 0.00}$ |
| MLP | $7.85_{\pm 0.35}$ | $1.91_{\pm 0.32}$ |
| MLP+RevIN+TB | $7.01_{\pm 0.08}$ | $1.48_{\pm 0.25}$ |
| RF+TB | $7.84_{\pm 0.04}$ | $2.40_{\pm 0.01}$ |
| FedFormer | $7.46_{\pm 0.04}$ | $1.29_{\pm 0.03}$ |
| LEM | $7.02_{\pm 0.04}$ | $1.42_{\pm 0.02}$ |
| VBS | $19.52_{\pm 0.00}$ | $5.60_{\pm 0.00}$ |
| KNF | $\mathbf{6.91}_{\pm 0.01}$ | $\mathbf{1.16}_{\pm 0.01}$ |
| KRNO | $\underline{6.95}_{\pm 0.16}$ | $\underline{1.25}_{\pm 0.05}$ |

## 4. Related work

The continuous time-shift operator introduced in this work is distinct from the Koopman operator (Koopman, 1931a), an infinite-dimensional linear operator acting on a space of observables, which has been extensively studied in dynamical systems and machine learning (Mezić, 2021; Brunton et al., 2022; Wang et al., 2023; Liu et al., 2023). The time-shift operator corresponding to a set of sufficiently smooth observables can be viewed as a continuous extension of the Koopman operator (Zhen et al., 2022b). This theoretical connection deserves further study.

Similar to neural ODE based methods (Chen et al., 2018; Rubanova et al., 2019; Kidger et al., 2020) and neural SDEs (Li et al., 2020a; Oh et al., 2024), our framework formulates time-series forecasting in a continuous setting. Our approach offers significant computational advantages during training, as it obviates the need for adjoint sensitivity methods to compute gradients of the loss function (similar to simulator-free approaches for learning SDEs, see, for example, Course & Nair (2023)). Furthermore, unlike typical NODE and SDE formulations that focus on temporal dynamics, our operator-theoretic framework naturally extends to spatio-temporal problems, inherently enabling super-resolution in both space and time.

Neural operators, such as the Fourier Neural Operator (FNO) (Li et al., 2020b) and its U-Net inspired variant U-shaped Neural Operator (*U-NO*) (Rahman et al., 2023), have advanced the learning of mappings between functions defined over spatio-temporal domains. Liu-Schiaffini et al. (2023) proposed FNO-based recurrent neural operator (RNO) architecture to address forecasting non-stationary dynamics on function spaces. As previously noted, Li et al. (2020b)

used the FNO to learn a map for the solutions of the Navier-Stokes equations from the time interval $[0, 10]$ to $(10, 50]$ and discussed the ability of this strategy to provide super-resolution in both space and time. This specific application can be viewed as an instance of our time-shift operator formalism with fixed time windows. Our framework is more general, employing learnable non-stationary kernels (via KRNOs) to parametrize the time-shift operator and treating the time window length as hyperparameters. Moreover, as demonstrated in our numerical studies (Section 3), KRNOs are considerably more parameter-efficient than FNOs while often achieving superior accuracy.

Transformer-based models have gained significant popularity in time-series forecasting (Zhang et al., 2024; Nie et al., 2023; Chen et al., 2021; Zhou et al., 2022; Wu et al., 2023; Liu et al., 2022; 2024). It is worth noting that most transformer architectures for time-series forecasting assume regularly sampled observations. A recent exception is T-PatchGNN (Zhang et al., 2024), designed for irregularly sampled data, against which we compare in Section 3.

## 5. Concluding remarks

We introduced a novel operator-theoretic framework for temporal and spatio-temporal forecasting based on learning a *continuous time-shift operator*. This paradigm offers a principled and robust approach to map the continuous history of a system to its future evolution, providing a continuous relaxation of traditional discrete-lag autoregressive models. To parameterize this operator, we proposed Khatri-Rao Neural Operators (KRNOs), a new architecture that leverages non-stationary integral transforms. KRNOs achieve nearly linear computational scaling with the number of quadrature points while allowing for exact kernel evaluations, a unique combination of efficiency and expressiveness.

The practical advantages of our approach include the ability to handle irregularly sampled observations, perform super-resolution forecasting in both space and time, and achieve significant parameter efficiency. Extensive numerical studies across 39 diverse test cases, encompassing physical simulations and challenging real-world time-series, demonstrate the efficacy and scalability of our method. KRNOs consistently achieve state-of-the-art or highly competitive performance, outperforming leading methods on several benchmarks, particularly for irregularly sampled data and complex spatio-temporal dynamics.

We envision several exciting research directions stemming from this work: further theoretical analysis of the time-shift operator itself, including its connections to other operator-theoretic constructs; development of advanced KRNO variants or alternative parameterizations; and the application of this paradigm to new domains.

## Acknowledgements

This work is supported by a Natural Sciences and Engineering Research Council of Canada Discovery Grant.

## Impact Statement

Work presented in this paper is mostly theoretical and computational in nature. At this stage, we don't see any potential societal impacts.

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

## A. Continuity of the time-shift operator for ODEs

**Lemma 1.** *Let $\dot{z}(\tau) = F(z(\tau))$, where $F : \mathbb{R}^n \to \mathbb{R}^n$ is Lipschitz continuous and $\tau \in [t_p, t_f]$. Then the time-shift operator $\mathcal{A}_{t_p}^{t,t_f} : L^2([t_p, t]; \mathbb{R}^n) \to L^2((t, t_f]; \mathbb{R}^n)$ is continuous in the $L^2$ sense, i.e., for $t_p < t < t_f$, there exists a constant $C > 0$ such that*

$$||\mathcal{A}_{t_p}^{t,t_f} z_1 - \mathcal{A}_{t_p}^{t,t_f} z_2||_{L^2((t,t_f];\mathbb{R}^n)} \leq C||z_1 - z_2||_{L^2([t_p,t];\mathbb{R}^n)},$$

*where $z_1$ and $z_2$ are trajectories of $\dot{z} = F(z)$ corresponding to different initial conditions.*

*Proof.* Let $e(\tau) = z_1(\tau) - z_2(\tau)$ with $\tau \in [t_p, t_f]$, where $z_1, z_2 \in L^2([t_p, t_f]; \mathbb{R}^n)$ are two ODE trajectories corresponding to different initial conditions. Then, we have

$$||\mathcal{A}_{t_p}^{t,t_f} z_1 - \mathcal{A}_{t_p}^{t,t_f} z_2||_{L^2((t,t_f];\mathbb{R}^n)} = ||z_1 - z_2||_{L^2((t,t_f];\mathbb{R}^n)} = ||e||_{L^2((t,t_f];\mathbb{R}^n)}. \tag{9}$$

Noting that $||\dot{e}||_2 = ||F(z_1) - F(z_2)||_2 \leq L_F ||e||_2$, where $L_F > 0$ is the Lipschitz constant of $F$, we have

$$\frac{d}{d\tau}||e(\tau)||_2^2 = 2e(\tau)^T \frac{de}{d\tau} \leq 2||e(\tau)||_2 \left\|\frac{de}{d\tau}\right\|_2 = 2L_F ||e(\tau)||_2^2, \ \tau \in [t_p, t_f]. \tag{10}$$

Applying Grönwall's lemma (Ames & Pachpatte, 1997) to the preceding inequality, we have

$$||e||_{L^2((t,t_f];\mathbb{R}^n)}^2 = \int_t^{t_f} ||e(\tau)||_2^2 d\tau \leq ||e(t_p)||_2^2 \int_t^{t_f} e^{2L_F(\tau - t_p)} d\tau. \tag{11}$$

Since the trajectories are continuous over $[t_p, t_f]$, it follows from the extreme value theorem that $||F(z(t))||_2$ is bounded over this interval which in turn implies that $\dot{z} \in L^\infty([t_p, t_f]; \mathbb{R}^n) \subset L^2([t_p, t_f]; \mathbb{R}^n)$. In addition, since $z$ is square integrable, we have $z \in H^1([t_p, t_f]; \mathbb{R}^n)$. Noting that $H^1([t_p, t_f]; \mathbb{R}^n) \hookrightarrow C([t_p, t_f]; \mathbb{R}^n)$ due to the Sobolev embedding theorem, there exists an embedding constant $C_1 > 0$ such that

$$\begin{aligned} ||e(t_p)||_2 &\leq ||e||_{L^\infty([t_p,t];\mathbb{R}^n)} \leq C_1 \left( ||e||_{L^2([t_p,t];\mathbb{R}^n)} + ||\dot{e}||_{L^2([t_p,t];\mathbb{R}^n)} \right) \\ &= C_1(1 + L_F)||e||_{L^2([t_p,t];\mathbb{R}^n)}. \end{aligned} \tag{12}$$

Using (9), (11), (12), we have

$$||\mathcal{A}_{t_p}^{t,t_f} z_1 - \mathcal{A}_{t_p}^{t,t_f} z_2||_{L^2((t,t_f];\mathbb{R}^n)} \leq C||z_1 - z_2||_{L^2([t_p,t];\mathbb{R}^n)},$$

where $C = \frac{C_1(1+L_F)}{\sqrt{2L_F}} \sqrt{\exp(2L_F(t_f - t_p)) - \exp(2L_F(t - t_p))}$. $\qquad \square$

The continuity of the spatio-temporal time-shift operator can be established for time-dependent PDEs under appropriate regularity assumptions. We leave this for future work.

**Time-shift operator for forced dynamical systems:** The notion of the continuous time-shift operator can be extended to forced dynamical systems. Consider a forced ODE of the form: $\dot{z}(\tau) = F(z(\tau), \xi(\tau))$, where $z(\tau) \in \mathbb{R}^n$ is the state, $\xi(\tau) \in \mathbb{R}^m$ is an exogenous forcing function (or control input), and $F : \mathbb{R}^n \times \mathbb{R}^m \to \mathbb{R}^n$ is assumed to be sufficiently regular. Similar to the unforced case, we can define a *forced time-shift operator*, denoted by $\mathcal{G}_{t_p}^{t,t_f}$, which maps the history of the state $z$ over a look-back window $[t_p, t]$ *and* the forcing function $\xi$ defined over the interval $[t_p, t_f]$ (which includes both look-back and prediction periods) to the future values of the state $z$ over the interval $(t, t_f]$. The forced time-shift operator $\mathcal{G}_{t_p}^{t,t_f}$ is therefore a map from $L^2([t_p, t]; \mathbb{R}^n) \times L^2([t_p, t_f]; \mathbb{R}^m)$ to $L^2((t, t_f]; \mathbb{R}^n)$, which we can define as follows:

$$z(\tau) = \left( \mathcal{G}_{t_p}^{t,t_f} \left( z|_{[t_p,t]}, \xi|_{[t_p,t_f]} \right) \right)(\tau), \quad \forall \tau \in (t, t_f], \tag{13}$$

where $z|_{[t_p,t]}$ and $\xi|_{[t_p,t_f]}$ denote the restriction of $z$ and $\xi$ to the intervals $[t_p, t]$ and $[t_p, t_f]$, respectively. When parametrizing the forced time-shift operator, it is critical to ensure *causality*, i.e., the predictions made at time instant $\tau$ are not influenced by the future values of the forcing function. This can be achieved by defining the integration domain of the kernel integral transform with respect to $\xi$ as $[t_p, \tau]$, where $\tau \in (t, t_f]$ denotes the time-stamp at which a prediction is being made. For example, a linear integral transform representation of the forced time-shift operator would take the form: $z(\tau) = \int_{t_p}^t \kappa_z(\tau, s) z(s) ds + \int_{t_p}^\tau \kappa_\xi(\tau, s) \xi(s) ds$. The notion of forced time-shift operators can be similarly extended to spatio-temporal systems, enabling the application of our operator-learning framework to systems with exogenous inputs.

# B. Proof for Proposition 2.1

We start by briefly clarifying what is meant by the quadrature nodes lying on a product grid. An example of a two-dimensional product grid is provided in Figure 6 below.

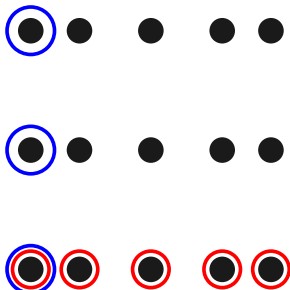

*Figure 6.* Example of a quadrature rule which lies on a product grid. Circled in blue are the quadrature nodes along the first dimension and circled in red are the quadrature nodes along the second dimension.

Proposition 1 states that if:

1. the quadrature nodes lie on a product grid, $X = \bar{t} \times x^{(1)} \times \ldots x^{(d-1)}$ where $\bar{t}, x^{(i)} \in \mathbb{R}^n$ indicate the quadrature nodes along the time dimension and the $i^{\text{th}}$ dimension of $x$ respectively such that $N = n^d$ (for the general case $N = \Pi_{i=1}^d n_i$) and

2. the kernel function has a component-wise product structure of the form given in Equation (7) reproduced below for clarity:

$$\kappa(\{t, x\}, \{t', x'\}) = \kappa^{(1)}(t, t') \odot \left( \odot_{i=2}^d \kappa^{(i)}([x]_{i-1}, [x]'_{i-1}) \right)$$

Then the kernel function evaluated at the quadrature nodes inherits the Khatri-Rao product structure provided in Equation (8) and reproduced below,

$$\kappa(X, X) = \kappa^{(1)}(\bar{t}, \bar{t}) * \left( \overset{d}{\underset{i=2}{*}} k^{(i)}(x^{(i-1)}, x^{(i-1)}) \right).$$

Here $\kappa^{(i)}(\cdot, \cdot) \in \mathbb{R}^{qn \times pn}$ is a block-partitioned matrix where block $jk$ is the $jk^{\text{th}}$ output from the component kernel $\kappa^{(i)}$ evaluated on the outer product of the quadrature nodes along the $i^{\text{th}}$ dimension.

*Proof.* We start by observing that $\kappa(X, X)$ can be block-partitioned into $q \times p$ blocks of size $N \times N$.

$$\kappa(X, X) = \begin{bmatrix} \kappa_{1,1} & \kappa_{1,2} & \ldots & \kappa_{1,p} \\ \kappa_{2,1} & \kappa_{2,2} & \ldots & \kappa_{2,p} \\ \vdots & & \ddots & \vdots \\ \kappa_{q,1} & \kappa_{q,2} & \ldots & \kappa_{q,p} \end{bmatrix}. \tag{14}$$

Each of these $N \times N$ blocks inherits the product structure of Equation (8),

$$\kappa_{j,k} = \odot_{i=1}^d \kappa_{j,k}^{(i)}(X[:, i-1], X[:, i-1]), \tag{15}$$

where $\kappa_{j,k}^{(i)}(X[:, i-1], X[:, i-1]) \in \mathbb{R}^{N \times N}$ is the $jk^{\text{th}}$ output of the $i^{\text{th}}$ component kernel function evaluated on the $i^{\text{th}}$ dimension of the quadrature nodes. The $jk^{\text{th}}$ block in the kernel evaluated at the quadrature nodes can be written as the Kronecker product as follows (Van Loan, 2000),

$$\kappa_{j,k} = \kappa_{j,k}^{(1)}(\bar{t}, \bar{t}) \otimes \left( \otimes_{i=2}^d \kappa_{j,k}^{(i)}(x^{(i-1)}, x^{(i-1)}) \right), \tag{16}$$

where $\kappa_{j,k}^{(i)}(x^{(i-1)}, x^{(i-1)}) \in \mathbb{R}^{n \times n}$, each $\bar{t}, x^{(i)} \in \mathbb{R}^n$ indicate the one-dimensional quadrature nodes along the time and $i^{\text{th}}$ dimension respectively. The Khatri-Rao product structure follows from substituting (16) into (14). $\qquad\square$

The above proof can be generalized to cases where the quadrature nodes are distributed on a product grid with a variable number of nodes along each dimension, i.e., $N = \Pi_{i=1}^{d} n_i$.

## C. Generalization of Proposition 2.1

In this section, we generalize Proposition 1 to the case where the input and output are defined over different spatio-temporal domains and different quadrature nodes are used for the input and output functions. We start by rewriting the kernel integral transform layer as a map with the input function $v_\ell : \Omega_\ell \times \mathcal{I}_\ell \to \mathbb{R}^p$ and the output function $v_{\ell+1} : \Omega_{\ell+1} \times \mathcal{I}_{\ell+1} \to \mathbb{R}^q$ as follows

$$v_{\ell+1}(t_{\ell+1}, x_{\ell+1}) = \mathcal{K}(v_\ell)(t_{\ell+1}, x_{\ell+1})$$
$$= \int_{\Omega_\ell} \int_{(0,\tau)} \kappa(\{t_{\ell+1}, x_{\ell+1}\}, \{t'_\ell, x'_\ell\})v_\ell(t'_\ell, x'_\ell)dt'_\ell dx'_\ell + W v_\ell(\Phi_\ell(t_{\ell+1}, x_{\ell+1})) + b,$$

where $\kappa : \mathbb{R} \times \Omega_\ell \times \mathbb{R} \times \Omega_{\ell+1} \to \mathbb{R}^{q \times p}$ is a matrix-valued kernel, $W \in \mathbb{R}^{q \times p}$ is a weight matrix, $\Phi_\ell : \Omega_{\ell+1} \to \Omega_\ell$ is a map between the output and input domains, and $b \in \mathbb{R}^q$ is a bias vector.

Let $X_\ell$ and $X_{\ell+1}$ denote the sets of quadrature nodes for the input and output domains, respectively. The quadrature nodes over the domain of the input function are assumed to lie on a product grid, i.e., $X_\ell = \bar{t}_\ell \times x_\ell^{(1)} \times \ldots x_\ell^{(d-1)}$, where $\bar{t}_\ell \in \mathbb{R}^{n_\ell}$ and $x_\ell^{(i)} \in \mathbb{R}^{n_\ell}$ denote the quadrature nodes along the input time dimension and the $i^{\text{th}}$ dimension of $x_\ell$, respectively, such that $N_\ell = n_\ell^d$ (for the general case when the number of quadrature nodes along the $i^{\text{th}}$ dimension is $n_{\ell_i}$, we have $N_\ell = \Pi_{i=1}^{d} n_{\ell_i}$). Similarly, the quadrature nodes over the output domain are assumed to lie on a product grid, $X_{\ell+1} = \bar{t}_{\ell+1} \times x_{\ell+1}^{(1)} \times \ldots x_{\ell+1}^{(d-1)}$ where $\bar{t}_{\ell+1} \in \mathbb{R}^{n_{\ell+1}}$ and $x_{\ell+1}^{(i)} \in \mathbb{R}^{n_{\ell+1}}$ denote the quadrature nodes along the output time dimension and the $i^{\text{th}}$ dimension of $x_{\ell+1}$, respectively, such that $N_{\ell+1} = n_{\ell+1}^d$ (for the general case with a different number of quadrature nodes along each dimension $N_{\ell+1} = \Pi_{i=1}^{d} n_{\ell+1_i}$). As before, we will consider a kernel with a component-wise product structure of the form given in Equation (7).

Similar to the previous proof, we start by observing that $\kappa(X_{\ell+1}, X_\ell)$ can be block-partitioned into $q \times p$ blocks of size $N_{\ell+1} \times N_\ell$, i.e.,

$$\kappa(X_{\ell+1}, X_\ell) = \begin{bmatrix} \kappa_{1,1} & \kappa_{1,2} & \ldots & \kappa_{1,p} \\ \kappa_{2,1} & \kappa_{2,2} & \ldots & \kappa_{2,p} \\ \vdots & & \ddots & \vdots \\ \kappa_{q,1} & \kappa_{q,2} & \ldots & \kappa_{q,p} \end{bmatrix}. \tag{17}$$

Each $N_{\ell+1} \times N_\ell$ block inherits the product structure, i.e., $\kappa_{j,k} = \odot_{i=1}^{d} \kappa_{j,k}^{(i)}(X_{\ell+1}[:, i-1], X_\ell[:, i-1])$, where $\kappa_{j,k}^{(i)}(X_{\ell+1}[:, i-1], X_\ell[:, i-1]) \in \mathbb{R}^{N_{\ell+1} \times N_\ell}$ is the $jk^{\text{th}}$ output of the $i^{\text{th}}$ component kernel function evaluated on the $i^{\text{th}}$ dimension of the quadrature nodes. The $jk^{\text{th}}$ block in the kernel evaluated at the quadrature nodes can be written as

$$\kappa_{j,k} = \kappa_{j,k}^{(1)}(\bar{t}_{\ell+1}, \bar{t}_\ell) \otimes \left( \otimes_{i=2}^{d} \kappa_{j,k}^{(i)}(x_{\ell+1}^{(i-1)}, x_\ell^{(i-1)}) \right), \tag{18}$$

where $\kappa_{j,k}^{(i)}(x_{\ell+1}^{(i-1)}, x_\ell^{(i-1)}) \in \mathbb{R}^{n_{\ell+1} \times n_\ell}$. Substituting (18) into (17), we have

$$\kappa(X_{\ell+1}, X_\ell) = \kappa^{(1)}(\bar{t}_{\ell+1}, \bar{t}_\ell) * \left( \mathop{\ast}_{i=2}^{d} k^{(i)}(x_{\ell+1}^{(i-1)}, x_\ell^{(i-1)}) \right), \tag{19}$$

where $\kappa^{(i)}(\cdot, \cdot) \in \mathbb{R}^{q n_{\ell+1} \times p n_\ell}$ is a block-partitioned matrix where block $jk$ is the $jk^{\text{th}}$ output from the component kernel $\kappa^{(i)}$ evaluated on the outer product of the quadrature nodes along the $i^{\text{th}}$ dimension.

It follows from this result that we retain the original computational complexity of the KRNO operator even in situations where the inputs and outputs are defined over different domains. In addition, this result provides the flexibility of designing memory-efficient multi-resolution neural operators, where the hidden layers operate on variable-resolution representations of the input function. This generalized KRNO integral transform layer can be viewed as a continuous analog of upsampling and downsampling layers used in convolutional neural networks.

## D. Algorithm for Khatri-Rao structured matrix-vector products

In this section, we present an algorithm to efficiently compute the matrix-vector product associated with the Khatri-Rao product structured matrix defined in (8), without the need to explicitly construct the full matrix of size $qN \times pN$.

Let $A \in \mathbb{R}^{qN \times pN}$ be a block structured matrix of the form,

$$
A = \begin{bmatrix} A_{1,1} & A_{1,2} & \dots & A_{1,p} \\ A_{2,1} & A_{2,2} & \dots & A_{2,p} \\ \vdots & & \ddots & \vdots \\ A_{q,1} & A_{q,2} & \dots & A_{q,p} \end{bmatrix} \tag{20}
$$

where each $A_{j,k} = \otimes_{i=1}^{d} A_{j,k}^{(i)}$ and $A_{j,k}^{(i)} \in \mathbb{R}^{n \times n}$. Assuming $q, p << N$, the computational complexity associated with the matrix-vector product $u = Av$ can be reduced from $\mathcal{O}(N^2)$ to $\mathcal{O}(N^{1+1/d})$. In addition, the memory requirements are also reduced from $\mathcal{O}(N^2)$ to $\mathcal{O}(N^{2/d} + N)$. An efficient PyTorch implementation outlining the steps is provided below.

```python
def khatri_rao_mmprod(
    A: list[Float[Tensor, "q p n1 n2"]], V: Float[Tensor, "pN batch"]
) -> Float[Tensor, "qN batch"]:
    d = len(A) # size of the product grid (# of kernel components)
    q, p, _, _ = A[0].shape
    pN, bs = V.shape
    X = V.reshape(p, -1, bs).transpose(-2, -1)
    for i in range(d):
        Gd = A[i].shape[-1]
        bs_prod = X.shape[:-1]
        X = X.reshape(*bs_prod, Gd, -1)
        Z = A[i].unsqueeze(-3) @ X
        X = Z.transpose(-2, -1).reshape(q, p, bs, -1)
    return X.sum(1).transpose(-2, -1).reshape(-1, bs)
```

We note that the above algorithm is applicable to Khatri-Rao product structured matrix, as defined in (19), where the inputs and outputs are defined over different spatio-temporal domains (with each domain using a different set of quadrature nodes).

## E. Details on KRNO parametrization

As mentioned in the paper, we parametrize each component-wise kernel, $\kappa^{(i)} : \mathbb{R} \times \mathbb{R} \to \mathbb{R}^{q \times p}$ by a neural network. All neural nets use skip connections and layer normalization (Ba et al., 2016). In addition, before passing an input into the component function, we apply an input transformation $\phi : \mathbb{R} \times \mathbb{R} \to \mathbb{R}^m$,

$$
\phi(t, t') = \frac{1}{\sqrt{2}} \cos\left(\begin{bmatrix} t & t' \end{bmatrix} \omega + \beta\right), \tag{21}
$$

where $\omega \in \mathbb{R}^{2 \times m}$ and $\beta \in \mathbb{R}^m$. Such input feature transforms were found to be beneficial in prior works (Kissas et al., 2022)

## F. Practical aspects of learning the time-shift operator

Consider an $n$-dimensional multivariate discrete time-series dataset $\{z_t\}_{t=0}^{T}$. This dataset is first converted into pairs of input and output sequences over two non-overlapping time intervals $[t_p, t]$ and $(t, t_f]$, where $0 \le t_p < t < t_f \le T$, for various time instances $t$. The input sequence, denoted by $U_p^t = \{z_t\}_{t=t_p}^{t}$, includes $z_t$ given at $P$ time steps within the look-back window $[t_p, t]$. The output sequence, denoted by $U_f^t = \{z_t\}_{t=t}^{t_f}$, contains values of $z_t$ given at $H$ time steps within the prediction window $(t, t_f]$. These pairs of sequences, for different values of $t$, are used to approximate the continuous time-shift operator $A_{t_p}^{t, t_f}$ using KRNO. For estimating the KRNO parameters, we minimize the loss function, $\frac{1}{M} \sum_{i=1}^{M} ||U_f^{t_i} - \mathcal{A}_{t_p}^{t_i, t_f} U_p^{t_i}||_{L^2((t_i, t_f]; \mathbb{R}^n)}$, where $M$ is the number of input-output sequence pairs. As discussed previously, $t_p$ and $t_f$ are hyperparameters of the time-shift operator which are chosen using cross-validation.

Similar to temporal datasets, we consider spatio-temporal data comprising discrete snapshots of spatial fields $\{u_t(x)\}_{t=0}^{T}$, where $x \in \Omega_g$ with $\Omega_g$ representing a spatial grid over $\Omega$. This dataset is converted into pairs of input and output sequences

of spatial fields, $U_p^t(x)$ and $U_f^t(x)$, over time intervals $[t_p, t]$ and $(t, t_f]$, where $0 \leq t_p < t < t_f \leq T$. The input sequence $U_p^t(x) = \{u_t(x)\}_{t=t_p}^t$ contains spatial fields corresponding to $P$ time steps within the look-back window $[t_p, t]$. The output sequence $U_f^t(x) = \{u_t(x)\}_{t=t}^{t_f}$ contains spatial fields over $H$ time steps within the prediction window $(t, t_f]$. These sequence pairs are used to learn the spatio-temporal time-shift operator (3) using KRNO by minimizing the loss function, $\frac{1}{M} \sum_{i=1}^M \|U_f^{t_i} - \mathcal{A}_{t_p}^{t_i,t_f} U_p^{t_i}\|_{L^2(\Omega) \times L^2((t_i,t_f])}$, where $M$ is the number of input-output sequence pairs.

*Table 8.* Summary of all datasets (along with corresponding test cases and error metrics) considered in our numerical experiments. This leads to a total of 39 different test cases or error metrics. The top five models are listed for each case. It can be seen that KRNO achieves top performance on 17/39 cases and top-3 performance on 30/39 cases.

| Dataset | Test case (or error metric) | Best | Second | Third | Fourth | Fifth |
|---|---|---|---|---|---|---|
| Darcy flow | $\epsilon_u$ | **KRNO** | FNO | POD-DeepONet | DeepONet | - |
| Hyper-elastic | $\epsilon_\sigma$ | **KRNO** | FNO | DeepONet | - | - |
| | $\epsilon_\rho$ | **KRNO** | FNO-3D | LOCA | - | - |
| Shallow water | $\epsilon_{v_1}$ | **KRNO** | FNO-3D | LOCA | - | - |
| | $\epsilon_{v_2}$ | **KRNO** | FNO-3D | LOCA | - | - |
| 2D spiral | Short traj. | **KRNO** | Latent-ODE | T-PATCHGNN | - | - |
| | Long traj. | **KRNO** | Latent-ODE | T-PATCHGNN | - | - |
| MIMIC | MSE | **KRNO** | T-PATCHGNN | StemGNN | Warpformer | GRU-D |
| | MAE | T-PATCHGNN | **KRNO** | GRU-D | Warpformer | StemGNN |
| USHCN | MSE | **KRNO** | T-PATCHGNN | Crossformer | Warpformer | Graph Wavenet |
| | MAE | **KRNO** | T-PATCHGNN | Graph Wavenet | Warpformer | Neural Flow |
| Human Activity | MSE | T-PATCHGNN | Warpformer | **KRNO** | Graph Wavenet | GRU-D |
| | MAE | T-PATCHGNN | Warpformer | Graph Wavenet | FOURIER/GNN | **KRNO** |
| | Regular | **KRNO** | Neural LSDE | Neural LNSDE | Neural GSDE | LEAP |
| MuJoCo | 30% | **KRNO** | Neural GSDE | Neural LNSDE | Neural GSDE | LEAP |
| | 50% | **KRNO** | Neural GSDE | Neural LNSDE | Neural GSDE | LEAP |
| | 70% | **KRNO** | Neural LSDE | Neural LNSDE | Neural GSDE | LEAP |
| | AirPassengers | LLaMA-2 | ARIMA | **KRNO** | GPT-3 | SM-GP |
| | AusBeer | N-BEATS | LLaMA-2 | GPT-3 | **KRNO** | ARIMA |
| | GasRateCO2 | SM-GP | **KRNO** | ARIMA | LLaMA-2 | N-BEATS |
| Darts | MonthlyMilk | GPT-3 | LLaMA-2 | **KRNO** | SM-GP | N-HiTS |
| | sunspots | **KRNO** | ARIMA | GPT-3 | LLaMA-2 | N-HiTS |
| | Wine | **KRNO** | GPT-3 | ARIMA | TCN | N-HiTS |
| | Wooly | N-HiTS | ARIMA | SM-GP | **KRNO** | GPT-3 |
| | HeartRate | TCN | GPT-3 | SM-GP | **KRNO** | N-HiTS |
| | Monthly | KNF | Nbeats-I+G | Smyl | Montero et al | **KRNO** |
| | Weekly | **KRNO** | KNF | Montero et al | Smyl | - |
| M4 | Daily | KNF | **KRNO** | Montero et al | Smyl | - |
| | Hourly | Smyl | KNF | Montero et al | **KRNO** | - |
| | Yearly | Nbeats-I+G | Smyl | Montero et al | KNF | **KRNO** |
| | Quarterly | Nbeats-I+G | Smyl | Montero et al | KNF | **KRNO** |
| | $(1 \sim 5)$ | MLP+RevIN+TB | KNF | **KRNO** | LEM | VARIMA |
| Crypto | $(6 \sim 10)$ | KNF | **KRNO** | MLP+RevIN+TB | LEM | FedFormer |
| | $(11 \sim 15)$ | KNF | **KRNO** | LEM | MLP+RevIN+TB | RF+TB |
| | $(1 \sim 15)$ | KNF | **KRNO** | MLP+RevIN+TB | LEM | FedFormer |
| | $(1 \sim 10)$ | VARIMA | KNF | **KRNO** | LEM | MLP+RevIN+TB |
| Player Traj | $(11 \sim 20)$ | KNF | VARIMA | FedFormer | **KRNO** | MLP+RevIN+TB |
| | $(21 \sim 30)$ | KNF | **KRNO** | FedFormer | VARIMA | LEM |
| | $(1 \sim 15)$ | KNF | **KRNO** | VARIMA | FedFormer | LEM |

## G. Comparison of training and inference costs of KRNO and FNO

In this section, we compare the computational cost and memory requirements of KRNO and FNO-3D for different spatial resolutions using the shallow water dataset. We utilized the default settings for KRNO described earlier, except for the quadrature grid. For all the spatial resolutions, we employed a quadrature grid of size $32 \times 32 \times 5$ in the hidden KRNO integral transform layers. It is important to note that for high-resolution spatial data, the memory requirements and computational cost are primarily driven by computations in the first and last integral transform layers, which contain the highest number of

*Table 9.* Comparison of training memory requirements and computational cost of KRNO and FNO-3D models on shallow water problem for different spatial resolutions (with a batch size of 8). Two FNO-3D configurations are compared: one with a fixed number of Fourier modes $m = 12$, and another with $m = \frac{S}{2} + 1$ that scales with the spatial resolution $S \times S$.

| Spatial resolution $S \times S$ | # Quadrature nodes | GPU memory (MB) | | | Time (seconds) | | |
|---|---|---|---|---|---|---|---|
| | | KRNO | FNO-3D $(m = 12)$ | FNO-3D $(m = \frac{S}{2}+1)$ | KRNO | FNO-3D $(m = 12)$ | FNO-3D $(m = \frac{S}{2}+1)$ |
| $32 \times 32$ | $5,120\,(N)$ | 1,390 | 854 | 1,074 | 0.0279 | 0.0216 | 0.0234 |
| $64 \times 64$ | $20,480\,(4N)$ | 2,776 | 1,350 | 2,772 | 0.0394 | 0.0252 | 0.0347 |
| $96 \times 96$ | $46,080\,(9N)$ | 4,884 | 2,124 | 5,264 | 0.0626 | 0.0405 | 0.0710 |
| $128 \times 128$ | $81,920\,(16N)$ | 7,608 | 3,318 | 8,994 | 0.0999 | 0.0704 | 0.1297 |
| $160 \times 160$ | $128,000\,(25N)$ | 10,040 | 4,872 | 13,764 | 0.1584 | 0.1114 | 0.2088 |

**Note:** $N = 32 \times 32 \times 5 = 5120$.

*Table 10.* Comparison of inference memory requirements and computational cost for KRNO and FNO-3D models on shallow water problem for different spatial resolutions (with a batch size of 8). Two FNO-3D configurations are compared: one with a fixed number of Fourier modes $m = 12$, and another with $m = \frac{S}{2} + 1$ that scales with the spatial resolution $S \times S$.

| Spatial resolution $S \times S$ | # Quadrature nodes | GPU memory (MB) | | | Time (seconds) | | |
|---|---|---|---|---|---|---|---|
| | | KRNO | FNO-3D $(m = 12)$ | FNO-3D $(m = \frac{S}{2}+1)$ | KRNO | FNO-3D $(m = 12)$ | FNO-3D $(m = \frac{S}{2}+1)$ |
| $32 \times 32$ | $5,120\,(N)$ | 708 | 942 | 1,074 | 0.0107 | 0.0065 | 0.0062 |
| $64 \times 64$ | $20,480\,(4N)$ | 1,314 | 1,294 | 2,772 | 0.0134 | 0.0069 | 0.0071 |
| $96 \times 96$ | $46,080\,(9N)$ | 2,366 | 1,622 | 5,264 | 0.0185 | 0.0108 | 0.0149 |
| $128 \times 128$ | $81,920\,(16N)$ | 3,796 | 2,428 | 8,994 | 0.0312 | 0.0210 | 0.0288 |
| $160 \times 160$ | $128,000\,(25N)$ | 5,644 | 3,292 | 11,288 | 0.0502 | 0.0315 | 0.0456 |

quadrature nodes. For the FNO-3D model, the number of Fourier modes ($m$) in each spatial dimension has a significant impact on the computational complexity. For instance, when training on data with spatial resolution $160 \times 160$, increasing $m$ from 12 to 81 increases the memory requirements from $4,872$ MB to $13,764$ MB and the training time per iteration from $0.1114$ seconds to $0.2088$ seconds. In addition, the parameter count of FNO-3D dramatically increases from $5.5$ million to $252$ million when $m$ increases from 12 to 81. In contrast, KRNO maintains a fixed parameter count of $145,319$, independent of the size of the quadrature grid and the resolution of the dataset.

The memory requirements and computational cost during training and inference are shown in Table 9, Table 10, and Figure 7. As discussed earlier, the memory requirements and time complexity of FNO-3D grow dramatically with an increase in the number of Fourier modes. We observe that the memory usage of KRNO is significantly higher during training than at inference. We believe that this can be reduced by further optimizing the implementation of Khatri-Rao matrix-vector products.

We would like to highlight that our current implementation uses a mid-point quadrature scheme for the temporal dimension and a trapezoidal quadrature scheme for spatial dimensions to evaluate the integral transform layers. Additionally, in our current implementation, the quadrature nodes are defined over the spatial mesh of the input function. While this approach is reasonable for the problems we are considering, it is not the most suitable (and tends to be overly conservative) for high-resolution spatio-temporal datasets. For such datasets, the memory requirements can be significantly reduced by using quadrature nodes that are defined over a lower-resolution spatial grid which is independent of the input's spatial resolution.

Alternatively, we could design a quadrature scheme that uses a low-resolution subsampling of the input as nodes and generates quadrature weights on-the-fly to meet a specified target precision. This would not only enhance accuracy and efficiency for high-resolution spatio-temporal datasets but also improve the performance of KRNO for irregularly spaced observations. We plan to explore this in future work.

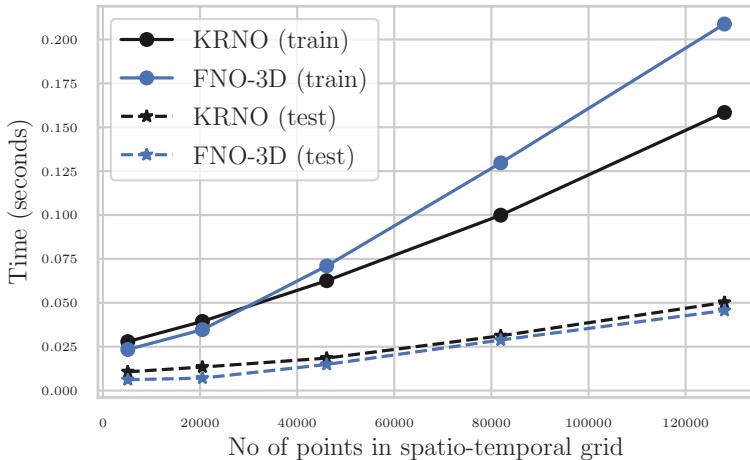

*Figure 7.* Training and inference time per iteration for KRNO and FNO-3D models on the shallow water problem, plotted as a function of the number of quadrature nodes. For the FNO-3D model, the number of Fourier modes is set to $m = \frac{S}{2} + 1$, where $S$ is the spatial resolution. See the discussion in Appendix G for more details.

## H. Additional experimental details

This section provides additional details on the experimental setup used to generate the results presented in the main text. The default KRNO architecture used in most of the experiments has 3 kernel integral layers with 20 channels each, lifting and projection layers are parametrized by MLPs with one hidden layer containing 128 hidden units, and the kernels in the integral transform layers are parametrized by MLPs with 3 hidden layers with 32 hidden units each. We used the SiLU (Elfwing et al., 2018) activation function as the nonlinearity. In each test case, we used the same input and output quadrature grids in each kernel integral layer except for the final layer. For problems involving high spatial or temporal resolutions, adopting lower-resolution quadrature grids within the internal kernel integral layers is recommended as an effective strategy to reduce computational costs. The hyperparameter settings used for Darts, M4, Crypto, and Player Trajectory datasets are summarized in Table 12 and Table 14. The AdamW (Loshchilov & Hutter, 2019) optimizer is used for training all the models. All the computations were carried out on a single Nvidia RTX 4090 with 24GB memory.

In all experiments, we treat the $t_p$ and $t_f$ as fixed hyperparameters. We would like to highlight here that further work is needed to explore the possibility of training a single model on a dataset containing input/output trajectories for different settings of $t_p$ and $t_f$. This would enable the possibility of learning a model that can predict the dynamics at different length scales.

### H.0.1. ADDITIONAL DETAILS ON MIMIC, USHCN AND HUMAN ACTIVITY BENCHMARKS

**MIMIC**  The MIMIC dataset (Johnson et al., 2016), a publicly available clinical database, contains electronic health records from critical care patients. After applying preprocessing steps outlined by Biloš et al. (2021); Zhang et al. (2024), we extracted time series data for 23,457 patients. Each time series encompasses 96 variables recorded during the initial 48 hours post-admission. The goal here is to forecast target values for the subsequent 24-hour period using the data from the first 24 hours as observations.

**USHCN**  The United States Historical Climatology Network (USHCN) dataset (Menne et al., 2015) is utilized to address the common challenge of missing data in climate research, which often arises from issues like sensor malfunctions or data acquisition errors. This dataset originally encompasses daily measurements of five climate variables (precipitation, snowfall, snow depth, minimum and maximum temperature) collected from numerous meteorological stations across the United States. Following preprocessing methodologies adopted from previous studies (De Brouwer et al., 2019b; Schirmer et al., 2022), a specific subset is used, focusing on $1,114$ stations and an observational period spanning four years, from 1996 to 2000. We followed the preprocessing steps described by Zhang et al. (2024), which resulted in $26,736$ multivariate time series. The primary task here is to forecast the climate conditions for the subsequent 24 months using the climate data from the preceding 24 months for each time series.

**Human Activity**  The Human Activity dataset contains 3D positional data from four sensors worn on the left ankle, right ankle, belt, and chest. This data, initially consisting of 12 irregularly measured variables from five individuals performing activities like walking and sitting, was preprocessed according to Zhang et al. (2024). This resulted in $5,400$ irregularly-sampled multivariate time series, each spanning $4,000$ milliseconds. For forecasting, we use the initial $3,000$ milliseconds of each irregularly sampled multivariate time-series as observed data to predict sensor positions for the subsequent $1,000$ milliseconds.

**KRNO architecture details:**  The MIMIC, USHCN, and Human Activity benchmarks involve irregularly sampled, multivariate time-series with missing observations. To accommodate the missing observations, we need to augment the KRNO architecture with a learnable transformation that maps the irregularly sampled observations with missing states to a feature vector defined over the union of all observation time-stamps across all variables/states. We experimented with different preprocessing architectures and found that the best performance was achieved by augmenting the KRNO architecture with a simple one-dimensional CNN preprocessing module that is shared across the variables (states). This is similar in spirit to approaches used in T-PATCHGNN (Zhang et al., 2024) and PatchTST (Nie et al., 2023), where all the model weights are shared across all the variables (states). It is worth noting here that in the present work only the CNN preprocessing module weights are shared across the states, while the KRNO architecture does not use weight sharing.

We take the union of all time-stamps corresponding to each variable (state) in a single trajectory, and create a binary mask to indicate the missing observations. The CNN module takes this augmented sequence for each variable (state) along with the corresponding mask (treated as a two-channel input) and transforms it to a single-channel feature vector defined over the union of all time-stamps. The transformed sequence of features extracted from the CNN preprocessing module along with the union of time-stamps are then used as inputs to the KRNO architecture. For all three datasets, we used a CNN preprocessing module composed of three layers, each employing a kernel size of 9 with stride length of 1 and padding of 4. The number of output channels for the three layers is set to 30, 20, and 1, respectively. We used the LeakyReLU (Maas et al., 2013) activation function as the nonlinearity. The KRNO architecture used is the default configuration described in Section H. The AdamW optimizer is used for training and the learning rate is set to $10^{-3}$. The training, validation and test splits are created using 60%, 20%, and 20% of the data, respectively.

Our current KRNO implementation for these datasets uses a batch size of 1 since the length of the transformed sequences for each trajectory is different. Further work is required to explore how this limitation can be overcome. One promising approach involves converting variable-length sequences into fixed-length feature vectors, analogous to the methodology in T-PATCHGNN (Zhang et al., 2024), thereby enabling their use as batched inputs for the KRNO model.

**Baseline models**  On irregular time-series benchmarks, we compare the performance of KRNO with numerous SOTA models, including DLinear (Zeng et al., 2023), TimesNet (Wu et al., 2023), PatchTST (Nie et al., 2023), Crossformer (Zhang & Yan, 2023), Graph Wavenet (Wu et al., 2019), MTCNN (Wu et al., 2020), StemGNN (Cao et al., 2020), CrossGNN (Huang et al., 2023), FOURIER/GNN (Yi et al., 2023), GRU-$\Delta$t, GRU-D (Che et al., 2018), SeFT (Horn et al., 2020), Rain-Drop (Zhang et al., 2022), Warpformer (Zhang et al., 2023), mTAND (Shukla & Marlin, 2021), CRU (Schirmer et al., 2022), Neural Flow (Biloš et al., 2021), T-PATCHGNN (Zhang et al., 2024), GRU-ODE (De Brouwer et al., 2019a), ODE-RNN, Latent-ODE (Rubanova et al., 2019), Augmented-ODE (Dupont et al., 2019), ACE-NODE (Jhin et al., 2021), NCDE (Kidger et al., 2020), ANCDE (Jhin et al., 2024), EXIT (Jhin et al., 2022), LEAP (Jhin et al., 2023), Neural SDE (Li et al., 2020a), Neural LSDE (Oh et al., 2024), Neural LNSDE (Oh et al., 2024), Neural GSDE (Oh et al., 2024).

### H.1. 2D spiral example

This section presents additional numerical results to evaluate the performance of KRNO on irregularly spaced time-series data obtained for the two-dimensional spiral test case from (Chen et al., 2018) and compare against T-PATCHGNN (Zhang et al., 2024) and Latent-ODE (Rubanova et al., 2019). First, we consider 10 irregularly spaced short training trajectories, which represent high levels of irregularity, alongside one equispaced training trajectory. The equispaced trajectory is generated by sampling the states at 100 evenly spaced time stamps over the short time interval $[0, 15.61]$ seconds. To generate training datasets with a high level of irregularity, 100 new time-stamps are obtained by adding random noise $\epsilon_t \sim \mathcal{U}[0, 0.156]$ to the time stamps of the equispaced trajectory. Subsequently, the training trajectory is obtained by sampling the states at the randomly perturbed time-stamps. This randomization procedure was repeated to create 10 different training datasets where the distribution of the time-stamps is highly irregular, making this test case more challenging. Similarly, we generated 10 irregularly spaced trajectories alongside one equispaced training trajectory over the longer time

interval $[0, 65.88]$ seconds.

We consider T-PATCHGNN (Zhang et al., 2024) and Latent-ODE (Rubanova et al., 2019) as baseline methods. We trained all models on both short and long trajectories. For models trained on short trajectories, performance is evaluated on a test dataset containing the future states at 18 uniformly spaced time stamps over the interval $(15.61, 18.45]$ seconds. While for models trained on long trajectories, performance is evaluated on a test dataset containing the future states at 18 uniformly spaced time stamps over the interval $(65.88, 68.72]$ seconds. All models are trained by fixing the length of the time window corresponding to the input and the output sequences to be 1.419 seconds. We did not tune the KRNO hyperparameters for this experiment. The KRNO architecture consisted of 128 channels in both the lifting and projection layers, with three kernel integral layers, each containing four channels.

The predictive performance of all models is compared in Table 11. When the training trajectories are equispaced, the performance of both Latent-ODE and T-PATCHGNN models is comparable, with KRNO outperforming them by a factor of 10. For the case of irregularly-spaced training data, we provide the statistics of test MSE over 10 different randomly sampled trajectories generated using the procedure described earlier. Table 11 shows that KRNO significantly outperforms both Latent-ODE and T-PATCHGNN models on this extrapolation task.

Sample predictions from models trained on short and long trajectories are shown in Figures 8, 9, respectively. KRNO is the only model that is able to consistently predict the future states on both short and long trajectories. Both Latent-ODE and T-PATCHGNN models revert to predicting the training data. In all cases, on average, Latent-ODE performs better than T-PATCHGNN. As shown in Figure 9, there is significant variance in the predictive performance of T-PATCHGNN across different irregularly spaced trajectories.

We note that reduction in predictive accuracy when the distribution of the time-stamps is highly irregular is influenced by the time intervals where the sampling frequency is low. For instance, lower sampling frequency in the time interval close to the testing time window has the most significant impact on predictive accuracy. In such situations, the hyperparameters $t_p$ and $t_f$ would need to be carefully selected to improve the predictive performance.

Another important factor that impacts the predictive accuracy for irregularly-spaced training observations is the quadrature scheme used to approximate the kernel integral transform layers. Our current implementation uses a trapezoidal quadrature rule which is not ideal when the training data is sampled at a highly irregular frequency. It is expected that by adopting a quadrature scheme that obtains weights on-the-fly for irregularly spaced data (while meeting a target precision), the accuracy can be improved further.

*Table 11.* Comparison of test MSE of models trained on one equispaced trajectory and 10 trajectories (mean/std) with irregularity in the distribution of the time-stamps.

| Train Trajectory | | Test MSE | | |
|---|---|---|---|---|
| Length | Type | KRNO | Latent-ODE | T-PATCHGNN |
| Short | Equispaced | **0.081** | 0.931 | 1.944 |
| | Irregular | **0.147 ± 0.06** | 0.929 ± 0.12 | 1.740 ± 0.40 |
| Long | Equispaced | **0.768** | 4.939 | 5.221 |
| | Irregular | **0.974 ± 0.68** | 6.828 ± 3.91 | 64.023 ± 58.9 |

## H.2. Darts benchmarks

For all the datasets in Darts, we used 60%-20%-20% as a train-validation-test split. We performed a grid search on the Darts datasets using the hyperparameters listed in Table 12 to find the optimal hyperparameters. Model selection was done based on the normalized mean absolute error (NMAE) (22) on the validation set. Since the available training data in the Darts dataset is not sufficient to train a deep network, we conducted weight decay tuning to determine the optimal weight decay value using the selected hyperparameters. This optimal weight decay value was then used to train the final model using both the training and validation data. This final model is used to obtain predictions by forecasting recursively until the end of the testing window, shown in Figure 10. The evaluation metric, normalized MAE (NMAE), is computed as follows

$$NMAE(\mathbf{y}, \hat{\mathbf{y}}) = \frac{MAE(\mathbf{y}, \hat{\mathbf{y}})}{\frac{1}{n}\sum_{i=1}^{n}|y_i|} = \frac{MAE(\mathbf{y}, \hat{\mathbf{y}})}{\text{mean}(|\mathbf{y}|)}, \tag{22}$$

where $\mathbf{y}$ and $\hat{\mathbf{y}}$ are the truth and predicted time series.

*Table 12.* Hyperparameter tuning ranges used for Darts dataset.

| Learning rate | Integral layer channels | Hidden units in kernel | Look-back window length | Prediction window length | ReVIN |
|---|---|---|---|---|---|
| [1e-3, 5e-3] | [5, 10, 32] | [32, 64] | 10 to 100 | 5 to 100 | [True, False] |

### H.3. M4 benchmarks

We utilized the training and test datasets from the M4 competition (Makridakis et al., 2020). For all M4 datasets, the last 10% of the data for each time series in the training data is used as validation data. The testing process involves forecasting for a specified time period (testing window length) for each seasonality. The testing window lengths for each seasonality are shown in the parentheses next to the seasonality in Table 13.

*Table 13.* Comparison of sMAPE from KRNO with other baseline methods on M4 datasets. Results with $(\cdot)^{\dagger}$ were taken from Wang et al. (2023).

| Method | Monthly(18) | Weekly(13) | Daily(14) | Hourly(48) | Yearly(6) | Quarterly(8) |
|---|---|---|---|---|---|---|
| Montero et al. (2020) | 12.639 | 7.625$^{\dagger}$ | 3.097$^{\dagger}$ | 11.506 | 13.528 | 9.733 |
| Smyl (2020) | 12.126 | 7.817$^{\dagger}$ | 3.170$^{\dagger}$ | **9.328** | 13.176 | 9.679 |
| Nbeats-I+G | 12.024 | - | - | - | **12.924** | **9.212** |
| KNF (Wang et al., 2023) | **11.930**$^{\dagger}$ | 7.254$^{\dagger}$ | **2.990**$^{\dagger}$ | 11.294 | 13.800 | 10.008 |
| KRNO(ours) | 13.432 | **6.934** | 3.086 | 11.686 | 14.302 | 10.503 |

### H.4. Crypto and Player Trajectory benchmarks

For these two datasets, we used the same train-test splits as Wang et al. (2023). Similar to the M4 datasets, 10% of the data corresponding to each time series in the training data is used for validation. Following (Wang et al., 2023), the prediction window lengths for the Crypto and Player Trajectory datasets were set to 15 and 30, respectively. Table 15 presents a comparison of prediction errors between KRNO and baseline methods.

For the Crypto dataset, Table 15 details the test prediction errors for time steps 1-5, 6-10, and 11-15, along with the total error across all 15 steps. Similarly, for the Player Trajectory dataset, Table 15 reports errors for time steps 1-10, 11-20, and 21-30, and the total error over all 30 steps.

*Table 14.* Hyperparameter ranges used in M4, Crypto, and Player Trajectory datasets.

| Learning rate | Integral layer channels | Hidden units in kernel | Look-back window length | Prediction window length | ReVIN |
|---|---|---|---|---|---|
| [1e-3, 5e-3] | [16, 32] | [32, 64] | 3 to 192 | 1 to 18 | [True, False] |

*Table 15.* Comparison of RMSE from KRNO method with other baseline methods on Crypto and Player Trajectory datasets.

| Model | Crypto (Weighted RMSE $10^{-3}$) | | | | Basketball Player Trajectory (RMSE) | | | |
| --- | --- | --- | --- | --- | --- | --- | --- | --- |
| | (1~5) | (6~10) | (11~15) | Total | (1~10) | (11~20) | (21~30) | Total |
| VARIMA | $6.09_{\pm0.00}$ | $8.83_{\pm0.00}$ | $10.74_{\pm0.00}$ | $8.76_{\pm0.00}$ | $\mathbf{0.22}_{\pm0.00}$ | $\underline{0.90}_{\pm0.00}$ | $1.98_{\pm0.00}$ | $1.26_{\pm0.00}$ |
| MLP | $6.68_{\pm1.53}$ | $7.95_{\pm0.33}$ | $8.64_{\pm0.55}$ | $7.85_{\pm0.35}$ | $0.73_{\pm0.20}$ | $1.64_{\pm0.31}$ | $2.77_{\pm0.42}$ | $1.91_{\pm0.32}$ |
| MLP+RevIN+TB | $\mathbf{5.03}_{\pm0.08}$ | $7.16_{\pm0.13}$ | $8.41_{\pm0.06}$ | $7.01_{\pm0.08}$ | $0.37_{\pm0.02}$ | $1.16_{\pm0.03}$ | $2.25_{\pm0.04}$ | $1.48_{\pm0.25}$ |
| RF+TB | $6.62_{\pm1.30}$ | $7.99_{\pm0.24}$ | $8.51_{\pm1.19}$ | $7.84_{\pm0.04}$ | $0.86_{\pm0.01}$ | $2.10_{\pm0.02}$ | $3.48_{\pm0.02}$ | $2.40_{\pm0.01}$ |
| FedFormer | $5.61_{\pm0.05}$ | $7.50_{\pm0.03}$ | $8.89_{\pm0.03}$ | $7.46_{\pm0.04}$ | $0.43_{\pm0.02}$ | $0.92_{\pm0.02}$ | $1.97_{\pm0.04}$ | $1.29_{\pm0.03}$ |
| LEM | $5.27_{\pm0.02}$ | $7.23_{\pm0.06}$ | $8.23_{\pm0.05}$ | $7.02_{\pm0.04}$ | $0.33_{\pm0.01}$ | $1.08_{\pm0.02}$ | $2.18_{\pm0.02}$ | $1.42_{\pm0.02}$ |
| VBS | $15.23_{\pm0.00}$ | $14.46_{\pm0.01}$ | $26.49_{\pm0.01}$ | $19.52_{\pm0.00}$ | $0.90_{\pm0.00}$ | $2.84_{\pm0.00}$ | $9.24_{\pm0.00}$ | $5.60_{\pm0.00}$ |
| KNF | $\underline{5.24}_{\pm0.00}$ | $\mathbf{7.03}_{\pm0.01}$ | $\mathbf{7.63}_{\pm0.01}$ | $\mathbf{6.91}_{\pm0.01}$ | $\underline{0.26}_{\pm0.01}$ | $\mathbf{0.84}_{\pm0.01}$ | $\mathbf{1.81}_{\pm0.01}$ | $\mathbf{1.16}_{\pm0.01}$ |
| KRNO | $5.27_{\pm0.27}$ | $\underline{7.07}_{\pm0.17}$ | $\underline{7.72}_{\pm0.1}$ | $\underline{6.95}_{\pm0.16}$ | $0.27_{\pm0.03}$ | $0.93_{\pm0.05}$ | $\underline{1.94}_{\pm0.07}$ | $\underline{1.25}_{\pm0.05}$ |

## H.5. Spatial modeling problems

### H.5.1. DARCY PROBLEM

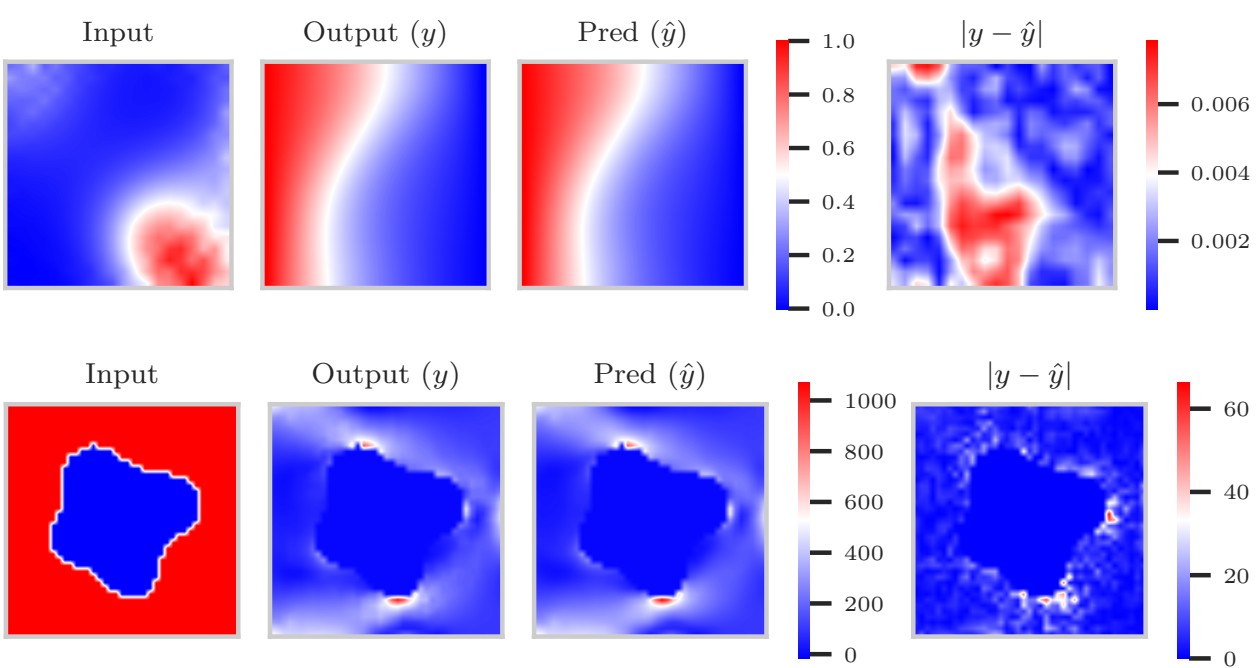

*Figure 11.* The top row presents a sample prediction from the test set for the Darcy-flow problem, while the bottom row illustrates a sample prediction for the elasticity problem.

We consider steady-state Darcy flow in a two-dimensional unit square domain $\Omega \in [0,1] \times [0,1]$, governed by the following PDE

$$-\nabla \cdot (a(\mathbf{x})\nabla u(\mathbf{x})) = f, \ \mathbf{x} \in \Omega, \ \text{where} \quad u(\mathbf{x}) = 0, \ \mathbf{x} \in \partial\Omega, \tag{23}$$

$a(\mathbf{x})$ is the permeability field, $u(\mathbf{x})$ is the pressure field, and $f$ is the source term. The goal here is to learn the pressure field $u(\mathbf{x})$ as a function of the permeability field $a(\mathbf{x})$. In this case, the permeability field is modelled as $a(\mathbf{x}) = \exp(F(\mathbf{x}))$, where $F(\mathbf{x})$ denotes a truncated Karhunen-Loéve (KL) expansion with 100 terms. We utilized the data provided in (Lu et al., 2022) for training and testing purposes. The training dataset consists of 1000 snapshots, and the testing data consists of 200 snapshots. The snapshots corresponding to both the input permeability field $a(\mathbf{x})$ and target pressure field $u(\mathbf{x})$ are

given on a uniform grid of size $20 \times 20$. We compared the relative test errors of KRNO with other operator-based methods such as DeepONet (Lu et al., 2021) and FNO (Li et al., 2020b). The results are presented in Table 2 and Figure 11.

The KRNO architecture used in the experiment has 3 kernel integral layers with 20 channels each, lifting and projection layers are parametrized by MLPs with one hidden layer containing 20 and 128 hidden units, respectively, and the kernels in the integral transform layers are parametrized by MLPs with 3 hidden layers with 128 hidden units each. We used the AdamW optimizer for training and the learning rate is set to $10^{-3}$.

### H.5.2. HYPER-ELASTIC PROBLEM

In this problem, we consider a hyper-elastic material within a unit domain $\Omega \in [0, 1] \times [0, 1]$ that contains a void of arbitrary shape at the center, as illustrated in the bottom row of Figure 11 (Li et al., 2023). The bottom edges of the unit cell are clamped, and we apply a tension traction of $t = [0, 100]$ to the top edge. The prior of the void radius is $r = 0.2 + \frac{0.2}{1+exp(\tilde{r})}$ with $\tilde{r} \sim \mathcal{N}(0, 4^2(\nabla + 3^2)^{-1})$, which satisfies the constraint $r \in [0.2, 0.4]$. The dynamics of an elastic material are represented by the following PDE

$$\rho\frac{\partial^2 \mathbf{u}}{\partial t^2} + \nabla \cdot \boldsymbol{\sigma} = 0 \tag{24}$$

where $\rho$ is the mass density, $u$ is the displacement vector, and $\boldsymbol{\sigma}$ is the stress tensor. Constitutive models are essential to define the behaviour of the system entirely. These models establish the relationship between the strain $\epsilon$ and stress tensors, effectively completing the mathematical framework necessary for solving the system. In this context, the material behaviour is characterized by the incompressible Rivlin-Saunders model, which uses energy density function parameters $C1 = 1.863 \times 10^5$ and $C2 = 9.79 \times 10^3$. We utilized the data provided in (Li et al., 2023) for both training and testing purposes. The training dataset consists of 1000 snapshots, and the testing data consists of 200 snapshots generated using a finite element solver.

The goal here is to learn the stress field as a function of the deformed mesh that corresponds to the void shape. We compared the relative test errors corresponding to predictions of our method with other ML-based operator methods such as FNO (Li et al., 2020b) and DeepONet (Lu et al., 2021). The results are presented in Table 2.

The KRNO architecture used in the experiment has 3 kernel integral layers with 20 channels each, lifting and projection layers are parametrized by MLPs with one hidden layer containing 128 hidden units, and the kernels in the integral transform layers are parametrized by MLPs with 3 hidden layers with 64 hidden units each. We used the AdamW optimizer for training and the learning rate is set to $10^{-3}$.

### H.6. Shallow water simulation

We provide some additional numerical results for the shallow water test case using the default KRNO architecture detailed in Section H. We followed the procedure described in Kovachki et al. (2023) in our numerical studies using FNO. Table 16 and Figure 12 compares results from different models trained for 200 epochs. The results presented show that FNO-3D performance improves when trained for 200 epochs. We also applied FNO-2D (Kovachki et al., 2023) to learn an autoregressive model that maps the spatio-temporal field at five time instants to the next time step. However, irrespective of the choice of hyperparameters, we were unable to learn an autoregressive model that provided stable predictions over the testing horizon. Due to this numerical issue, the FNO-2D model is only trained for 100 epochs. Representative predictions from all the models are shown in Figures 14, 15, 16, 17.

### H.7. Climate modeling example

The KRNO architecture used in the experiment has 3 kernel integral layers with 20 channels each, lifting and projection layers are parametrized by MLPs with one hidden layer containing 20 and 256 hidden units, respectively, and the kernels in the integral transform layers are parametrized by MLPs with 2 hidden layers with 64 hidden units each. We used the AdamW optimizer for training and the learning rate is set to $10^{-3}$.

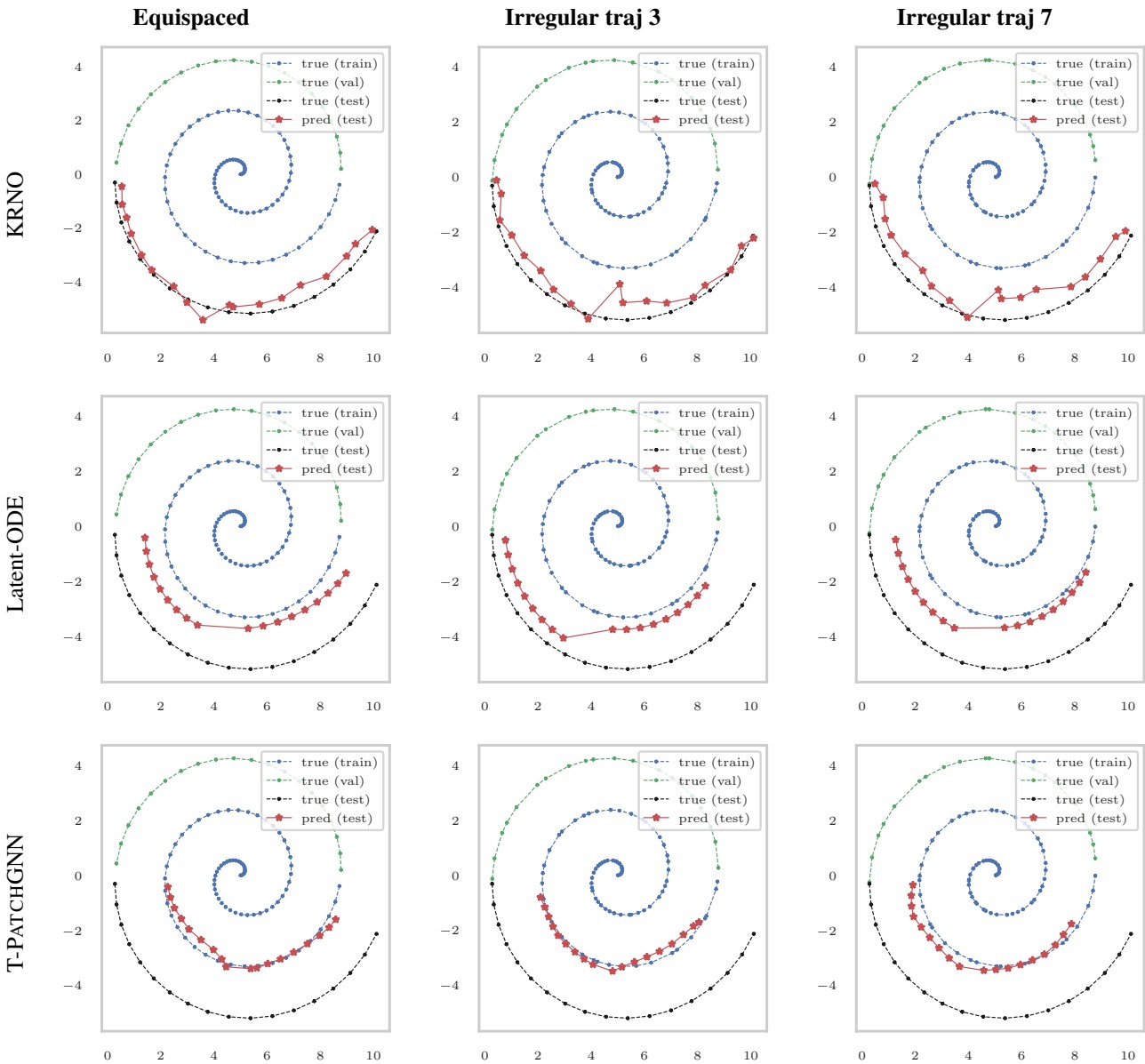

*Figure 8.* Comparison of forecasted predictions of KRNO with other models trained on short trajectories with equispaced and high levels of irregularity in the time-stamps.

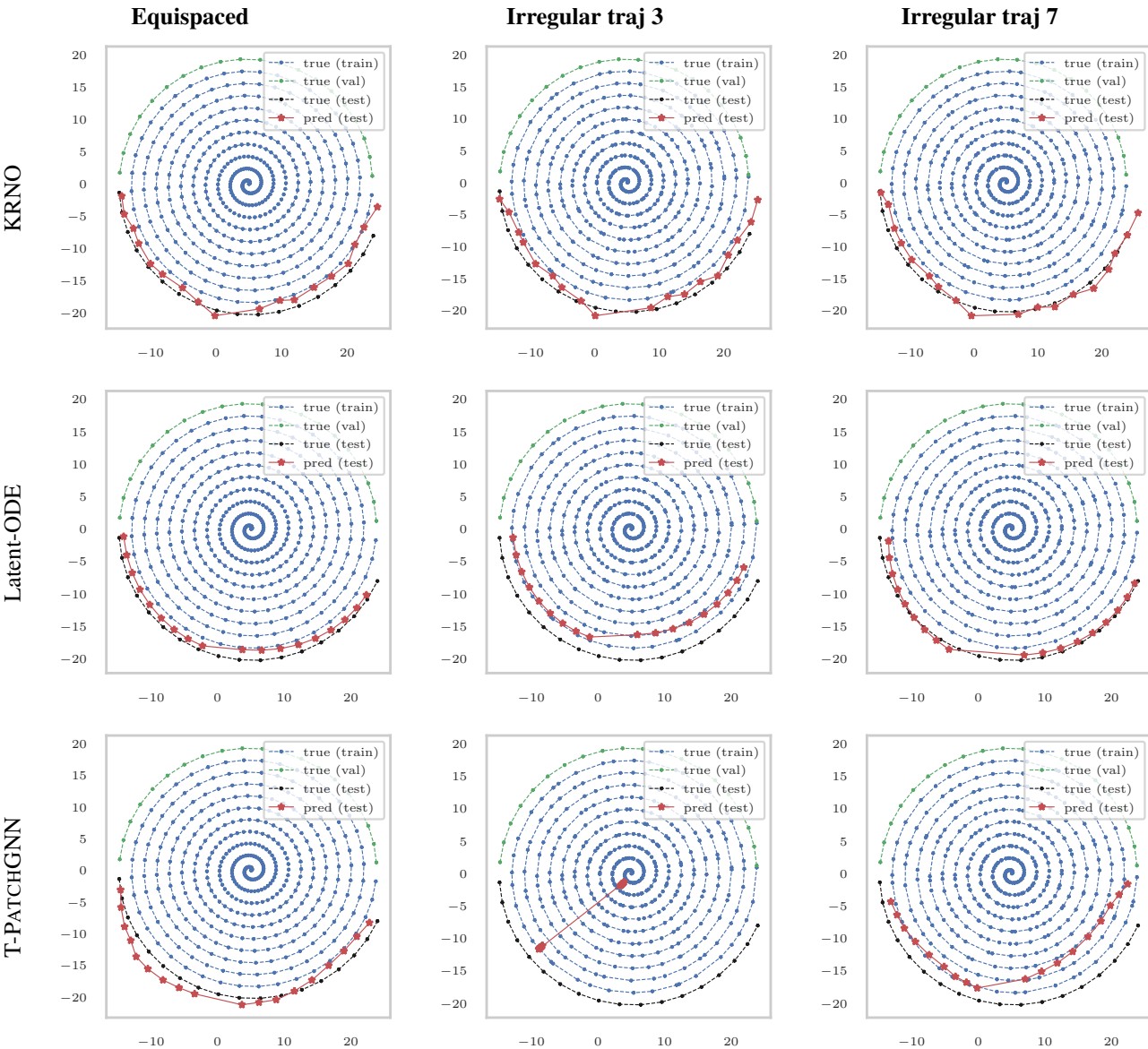

*Figure 9.* Comparison of forecasted predictions of KRNO with other models trained on long trajectories with equispaced and high levels of irregularity in the time-stamps.

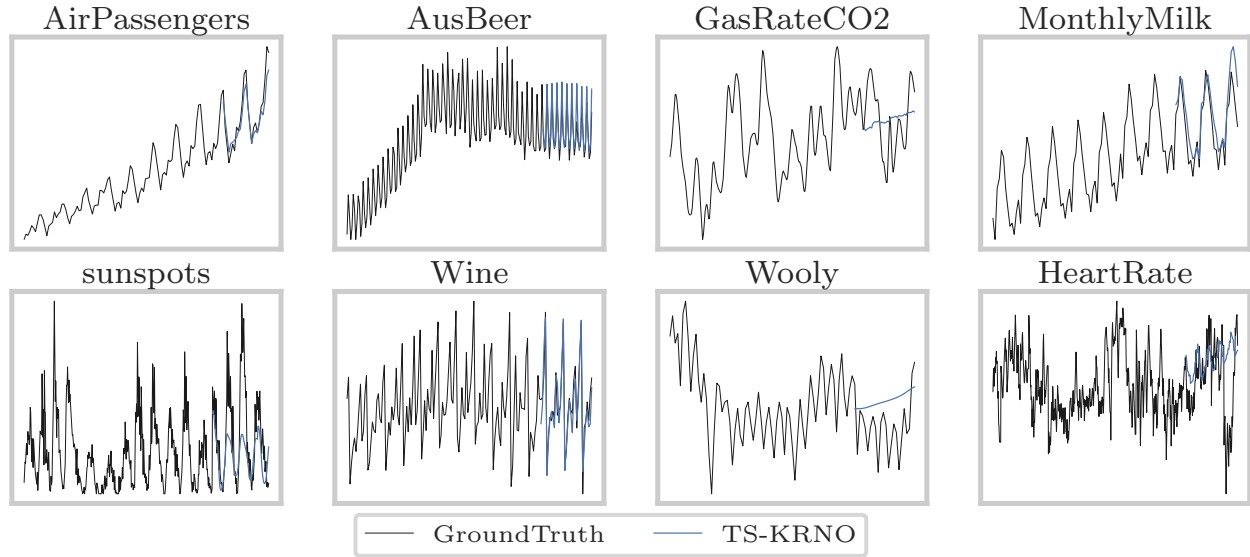

*Figure 10.* KRNO predictions on the Darts datasets

*Table 16.* Comparison of the average relative $L^2$ errors on the shallow water problem for the three field variables when training is conducted for 200 epochs.

| Method | $L^2$ relative error | | |
|--------|------|------|------|
| | $\rho$ | $u$ | $v$ |
| FNO-3D | 0.000719 | 0.01951 | **0.01174** |
| LOCA | 0.003091 | 0.15179 | 0.14942 |
| KRNO | **0.000331** | **0.01339** | 0.01406 |

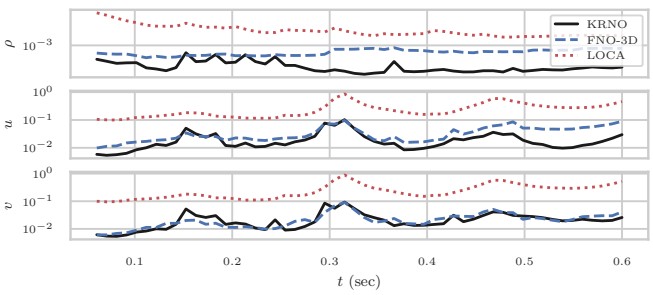

*Figure 12.* Comparison of the average relative $L^2$ errors as a function of time for the three field variables (across the 1000 test simulations) obtained using KRNO, FNO-3D and LOCA models trained for 200 epochs.

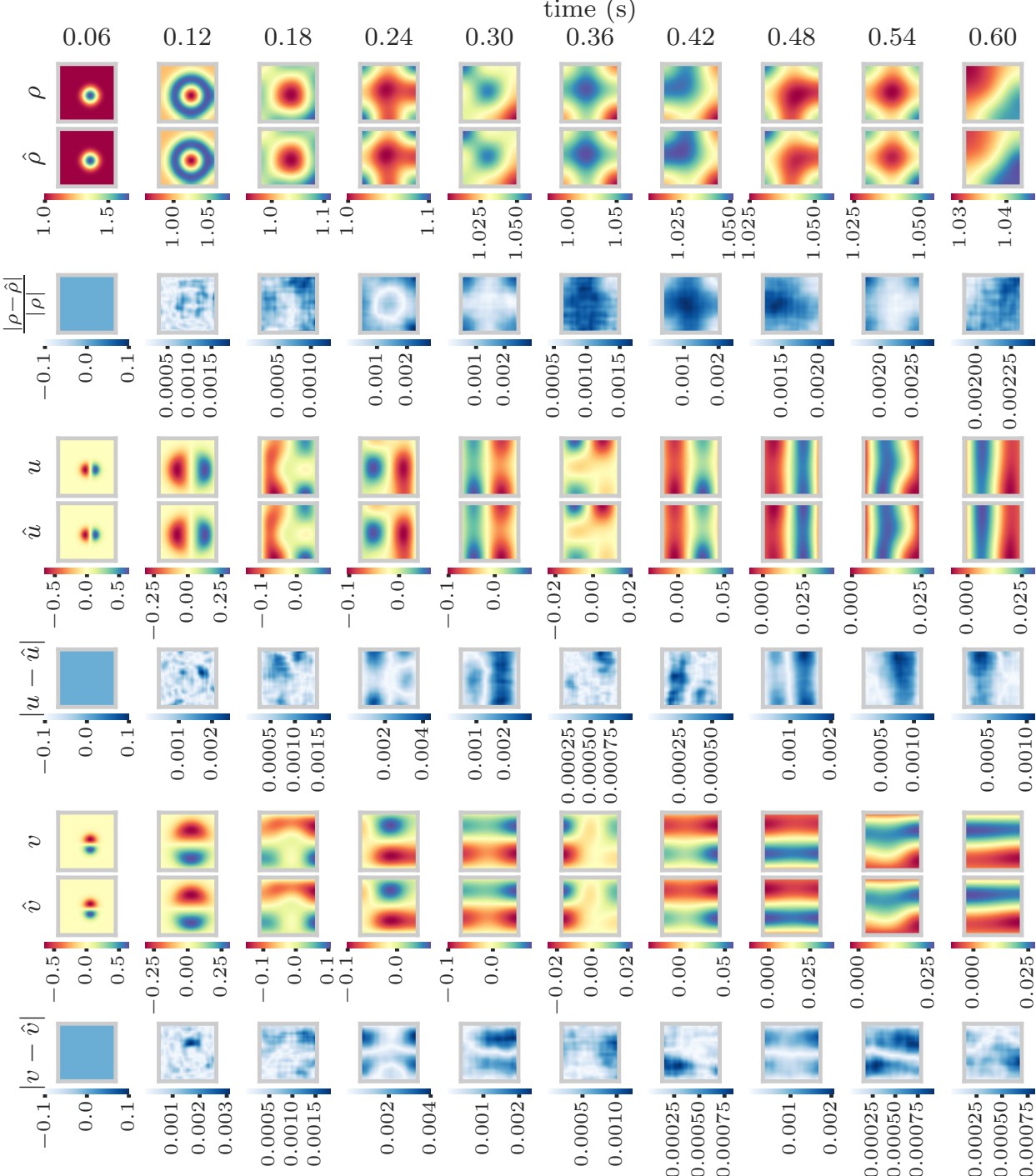

*Figure 13.* Shallow water problem: Figure shows the predictions $(\hat{\rho}, \hat{u}, \hat{v})$ for the three field variables along with the true fields $(\rho, u, v)$ as a function of time for a test simulation using KRNO trained for 100 epochs.

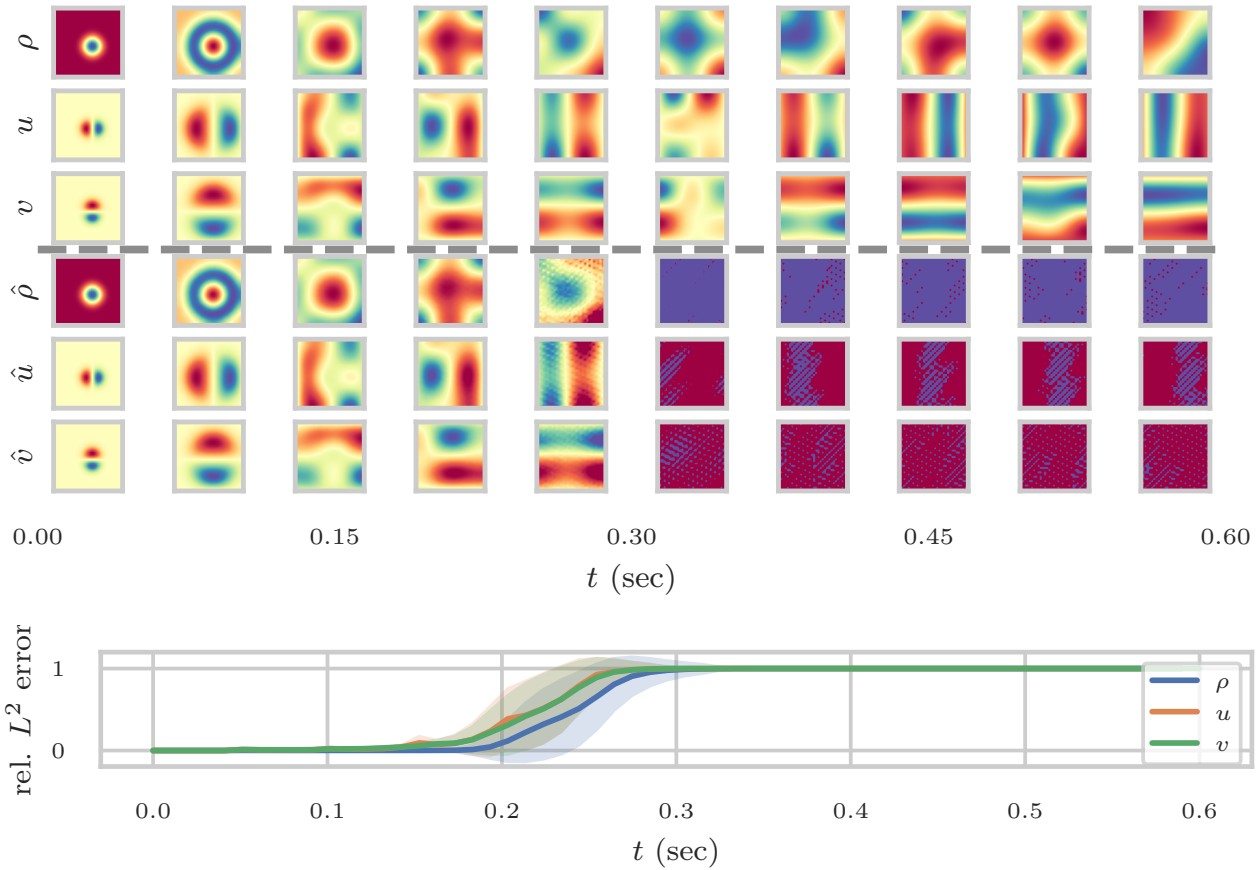

*Figure 14.* Shallow water problem: Top figure shows the predictions $(\hat{\rho}, \hat{u}, \hat{v})$ for the three field variables along with the true fields $(\rho, u, v)$ as a function of time for a test simulation using FNO-2D trained for 100 epochs. The bottom figure shows the error bars representing the $L^2$ relative errors for three field variables across the 1000 test simulations, with the shaded region indicating $\pm 1$ standard deviation.

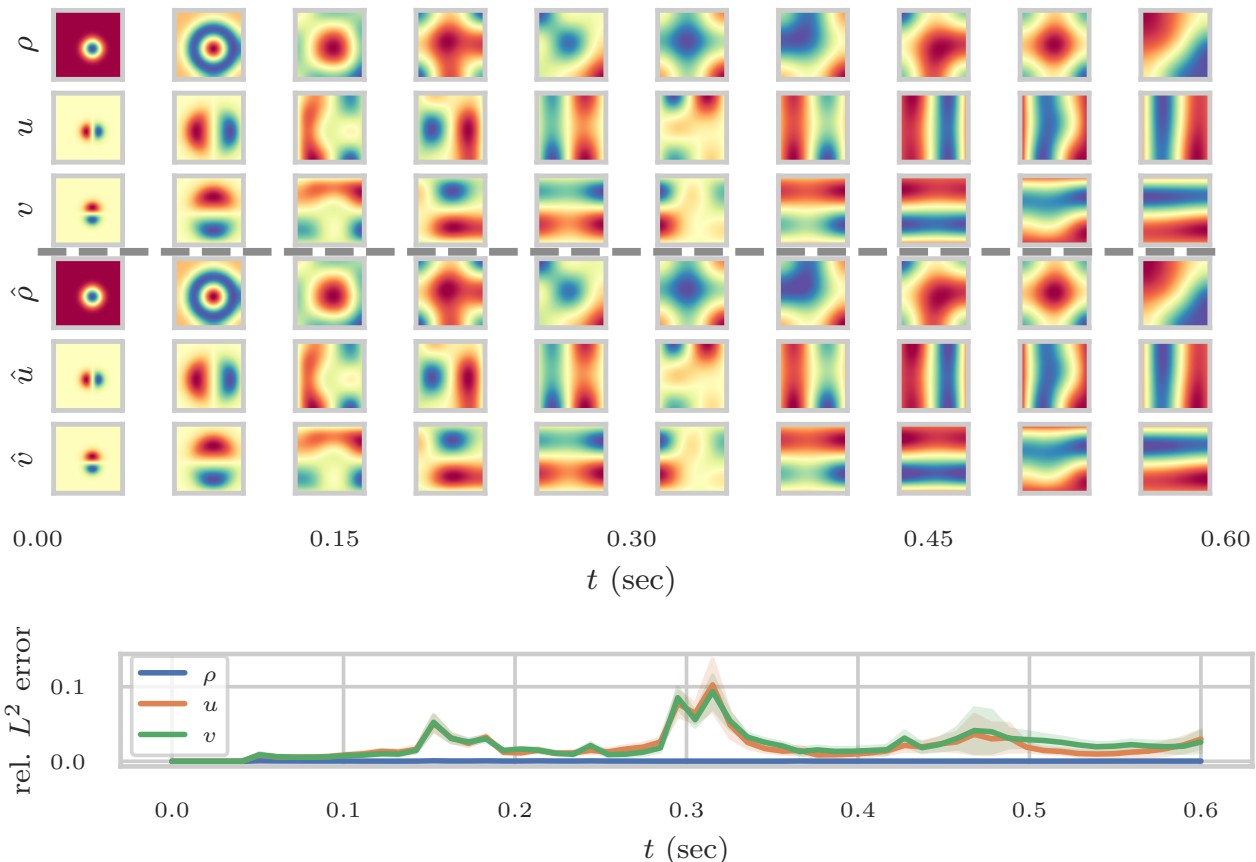

*Figure 15.* Shallow water problem: Top figure shows the predictions $(\hat{\rho}, \hat{u}, \hat{v})$ for the three field variables along with the true fields $(\rho, u, v)$ as a function of time for a test simulation using KRNO trained for 200 epochs. The bottom figure shows the error bars representing the $L^2$ relative errors for three field variables across the 1000 test simulations, with the shaded region indicating $\pm 1$ standard deviation.

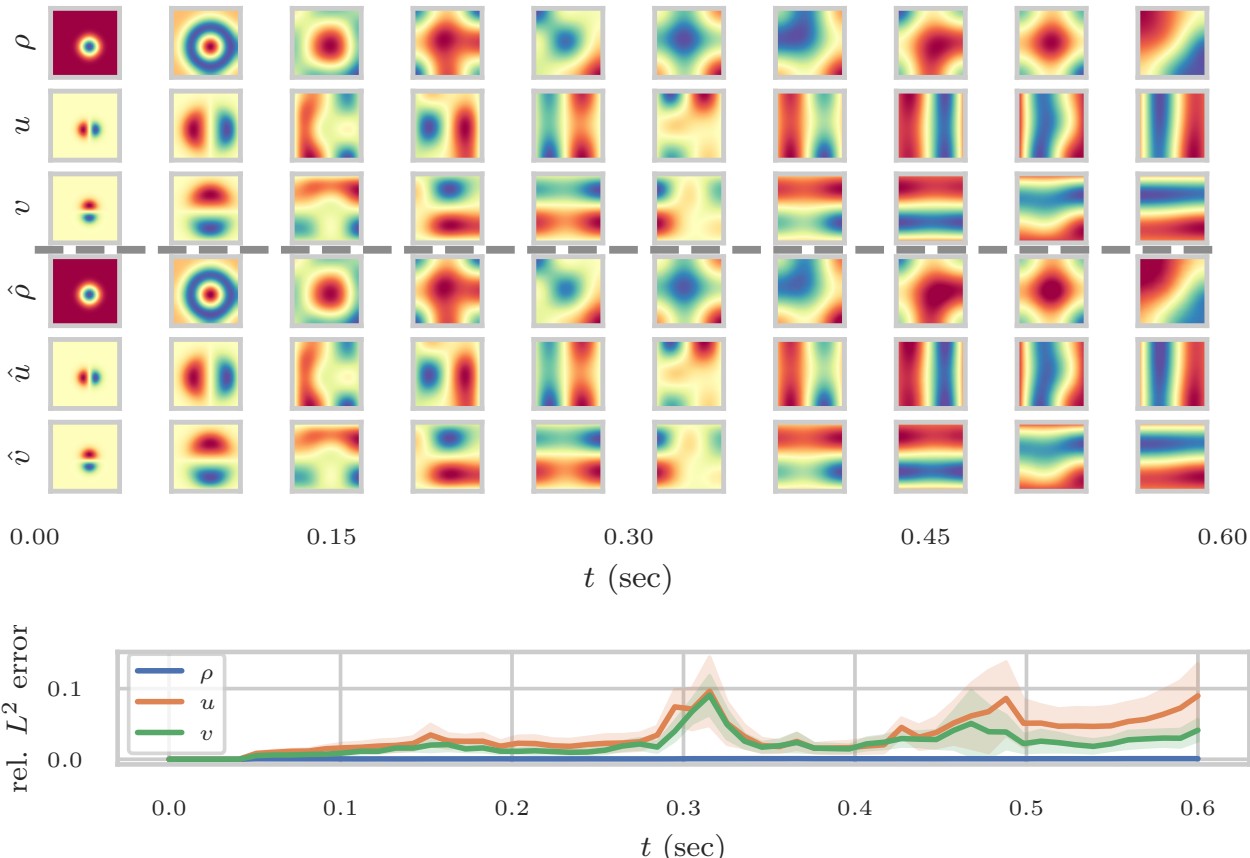

*Figure 16.* Shallow water problem: Top figure shows the predictions $(\hat{\rho}, \hat{u}, \hat{v})$ for the three field variables along with the true fields $(\rho, u, v)$ as a function of time for a test simulation using FNO-3D trained for 200 epochs. The bottom figure shows the error bars representing the $L^2$ relative errors for three field variables across the 1000 test simulations, with the shaded region indicating $\pm 1$ standard deviation.

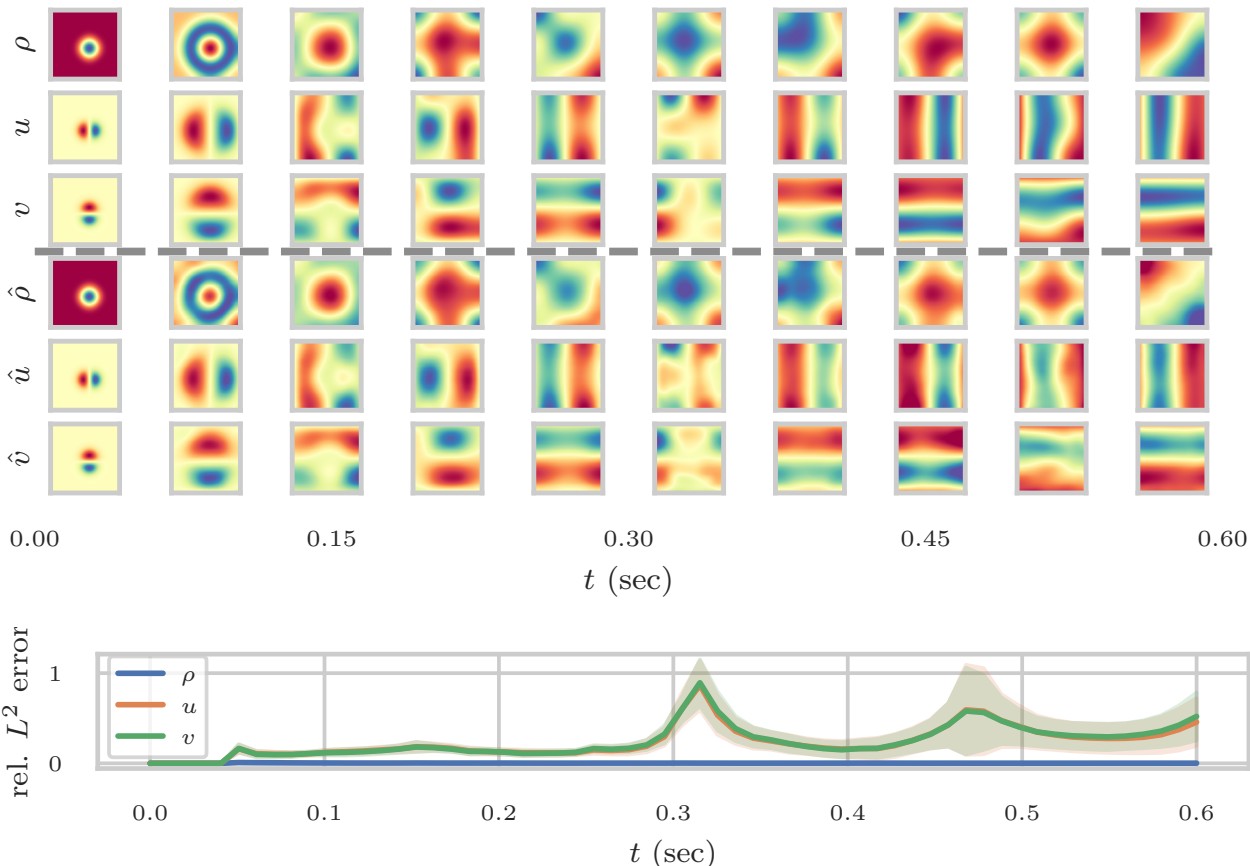

*Figure 17.* Shallow water problem: Top figure shows the predictions $(\hat{\rho}, \hat{u}, \hat{v})$ for the three field variables along with the true fields $(\rho, u, v)$ as a function of time for a test simulation using LOCA model trained for 200 epochs. The bottom figure shows the error bars representing the $L^2$ relative errors for three field variables across the 1000 test simulations, with the shaded region indicating $\pm 1$ standard deviation.

