# OpenReview forum: "Shifting Time: Time-series Forecasting with Khatri-Rao Neural Operators"
_ICML.cc/2025/Conference — ICML 2025 poster_

### Official Review · Reviewer_kNx6 · 2025-03-05

**Overall Recommendation:** 2

**Summary:**

The authors propose a time series forecasting method, leveraging continuous time-shift operators, which act as continuous analogs of the lag factor of the discrete-time autoregressive models or the upsampling/downsampling layers in CNNs, mapping the history of values up to an observation to a future window. To tackle the operator-learning problem, the authors introduce Khatri-Rao neural operators (KRNO) that define non-stationary integral-based transforms with the almost linear cost for spatio-temporal problems. The proposed method is evaluated on several real-world and synthetic spatiotemporal and standard temporal forecasting datasets, showcasing competitive performance with other popular time series methods. An anonymized code repository is also provided for reproducibility purposes.

## update after rebuttal

Assessing the overall impact of the contribution, including the theoretical results and experimental evaluation, and following the author's rebuttal that showcased incremental performance results on real-world irregularly sampled datasets (or without proven statistical significance), I maintain my initial ratings. During the rebuttal, the authors have not adequately addressed significant questions concerning the positioning of the contribution (W1, W2) or presented a thorough computational (e.g., time cost) analysis between the proposed method and considered baselines for the chosen tasks (which is a significant aspect in case of borderline performance improvements). Performance percentage improvements in terms of MSE are misleading (referring to the 2nd decimal with overlapping stds), and the proposed method is outperformed in Tables 5 and 6, while results during the rebuttal on irregular datasets showcase average improvements ranging from 1% to 7% for 2 out of the 3 datasets.

**Claims And Evidence:**

A few problematic claims:
- *(Almost) Linear Complexity Competitive to Operator-Based Methods:* The almost linear complexity of the method compared to other operator-based learning frameworks is not showcased in the main paper but rather in the appendix, where experimental results show that the proposed KRNO method is the worst in terms of train-test times per iteration for the spatiotemporal shallow water dataset (surpassed in several cases by the FNO method that has O(nlogn) complexity).
- *Continuous Analog of Discrete-Time Autoregressive Models:* It is not very clear how the proposed KRNO method is a continuous analog of the lag factor of the discrete-time autoregressive models, and for which selection of kernels this holds.

**Essential References Not Discussed:**

Based on the review comment above (**"Methods And Evaluation Criteria"**), several baseline methods for irregularly-sampled temporal forecasting are not discussed, including the recent SOTAs based on Neural SDEs (several methods are mentioned in [1]). For regular temporal forecasting, a more standard benchmark followed in the community is based on Time Series Library [2,3] (methods such as TimesNet, TimeXer, iTransformer, and PatchTST are first in ranking yet not tested in the paper).

[1] Oh, Y., Lim, D. Y., & Kim, S. (2024). Stable neural stochastic differential equations in analyzing irregular time series data. arXiv preprint arXiv:2402.14989.

[2] Time Series Library https://github.com/thuml/Time-Series-Library

[3] Wang, Y., Wu, H., Dong, J., Liu, Y., Long, M., & Wang, J. (2024). Deep time series models: A comprehensive survey and benchmark. arXiv preprint arXiv:2407.13278.

**Experimental Designs Or Analyses:**

The authors follow standard experimental designs for spatiotemporal and temporal forecasting (including optimization, metrics, and datasets). It is unclear if they measure the variance of models' performances with multiple runs with different random seeds if the results refer to a single fixed seed.

**Methods And Evaluation Criteria:**

Proposed methods and evaluation criteria (e.g., datasets) are generally appropriate for the task at hand but could be significantly enhanced to enable a thorough experimental evaluation.

**Baseline Methods:** Particularly for the case of irregularly sampled time series, only two baselines are used. More recent state-of-the-art methods can be found here [2].

**Datasets:** The proposed method is evaluated on two spatiotemporal forecasting datasets, including data for shallow water simulation and climate modeling, and several temporal forecasting datasets, including the M4 competition archive, the Darts archive, and the two additional crypto and player trajectory datasets. The datasets are commonly used in the (at least for the temporal) forecasting community. Yet several common datasets are missing in terms of regularly sampled time series (ETT, Electricity, Traffic etc, summarized here https://github.com/thuml/Time-Series-Library). The authors also generate an irregularly sampled synthetic dataset to validate their method for data with non-equidistant observation times. The synthetic 2D spiral dataset is rather limiting for the irregularly sampled time series. More prominent methods perform forecasting on apriori irregularly-sampled data or randomly downsample regular ones and do extrapolation (see [1,2]).

[1] Rubanova, Y., Chen, R. T., & Duvenaud, D. K. (2019). Latent ordinary differential equations for irregularly-sampled time series. Advances in neural information processing systems, 32.

[2] Oh, Y., Lim, D. Y., & Kim, S. (2024). Stable neural stochastic differential equations in analyzing irregular time series data. arXiv preprint arXiv:2402.14989.

**Other Comments Or Suggestions:**

No comments

**Other Strengths And Weaknesses:**

Summarized **strengths** of the paper:
- **[S1]** The authors propose a thoroughly presented operation-based theoretical framework that aims to tackle real-world spatiotemporal problems beyond numerical solvers.
- **[S2]** Broad tasks, such as temporal forecasting and irregular sampling, are approached.
- **[S3]** The included visualizations enhance the readability of the work.

Summarized **weaknesses** of the proposed study:
- **[W1]** The work is poorly presented compared with the neural-operator-based related works in the literature (such as FNO). Such related methods are abstractly mentioned in the introduction, and several details are given in the methods section, but very few are in the related work section. I suggest restructuring the related method's details throughout the text to highlight the proposed method’s contributions.
- **[W2]** The related work section is misplaced and limited regarding references. Only a few methods (Koopman, FNO) are explained in terms of operator-based learning, and one method is explained for irregularly sampled time series modeling (NeuralODEs). Baseline methods used in experiments should be explained in more detail along with more recent methods (see relevant review section above).
- **[W3]** Several mixed benchmarks in terms of datasets but with important state-of-the-art methods missing, especially for temporal forecasting and irregularly-sampled temporal modeling (see relevant review section above).
- **[W4]** The proposed method is, in several cases, outperformed by baselines, particularly for the temporal forecasting datasets. However, the performance significance of results is not justified if studied, e.g., were multiple runs with random seeds followed for all datasets and baselines? In some cases, the results for the baseline are directly taken from the relevant papers.
- **[W5]** The experiments on irregularly sampled time series are limited to simple synthetically generated data (2D spiral), which raises questions about the method's efficacy in real-world setups with irregular timestamps.
- **[W6]** Studies on the computational complexity of the proposed KRNO methods compared to SOTAs for temporal forecasting (beyond operator-based methods) are missing. This is essential to support the applicability of the process in practical scenarios with vast time series datasets, where simple and lightweight architectures are, in several cases, more favorable.

**Questions For Authors:**

1. **[Q1]:** Could authors improve the presentation and structuring of the related work (based on **[W1], [W2]**)?
2. **[Q2]:** Could you tackle the issues raised on experimental evaluation in terms of temporal SOTAs, methods applied to irregularly sampled data, and real-world irregularly sampled datasets (based on **[W3], [W4], [W5]**)?
3.  **[Q3]:** To what extent are neural operators for time series forecasting constrained to data exhibiting clear structures, smooth evolution, and spatial correlations? (for instance, NeuralODEs solve irregular sampling problems for specific physical datasets/are not successfully extended to standard forecasting)

**Relation To Broader Scientific Literature:**

This work is related to the problem of time series forecasting, including spatiotemporal and temporal settings, also extending to irregularly sampled time series that are very common in several engineering and scientific domains. It explores operator-based learning methods proposed for solving complex problems in spatiotemporal data and PDEs as an alternative to the complexity of numerical solvers. The proposed method introduces an efficient time-shift operator for capturing multiple levels of granularities and irregular sampling without the need to approximate the kernel while maintaining linear complexity.

**Theoretical Claims:**

Proofs for the properties of the time-shift operator and Proposition 2.1 have been checked for correctness. No issues detected.

---

> ### Author Rebuttal · Authors · 2025-04-01
>
> Thank you for the detailed review and valuable comments.
>
> ## 1. Claims and Evidence:
> >**(Almost) Linear Complexity..**
>
> Please refer to our first response to reviewer **dfXC** for more clarification on the computational complexity and runtime analysis of KRNO compared to FNO.
>
> >**Continuous Analog..**
>
> Using the proposed continuous time-shift operator, we learn an operator that maps the history of the dynamics over a past time-window into its future values over a subsequent time-window. This enables us to learn from irregularly sampled observations and to forecast at any given time over a fixed time-window, making it a continuous analog of the lag factor of the discrete-time autoregressive models. It is also worth noting that the continuity constant of the time-shift operator can be bounded in terms of the Lipschitz constants of the integral transform layers. We plan to study this in future work to show robustness to distribution shifts similar to the seminal work of Oh et al. (2024) on stable NSDEs.
>
>
> ## 2. Baselines and Evaluation Criteria:
> We agree that numerical studies on regularly sampled time-series do not allow a clear demonstration of the full capabilities offered by KRNO. We now include additional experiments on challenging irregularly-sampled time series benchmark datasets (MIMIC, USHCN, Human Activity, and MuJoCo) and the results are compared against a range of alternative approaches with SOTA performance such as T-PatchGNN, NeuralSDE, NeuralCDE, PatchTST, and Latent-ODE. KRNO achieves new SOTA performance on three of the four benchmarks.
>
> The results for the MuJoCo benchmark are provided below where we compare against the methods in the ICLR 2024 paper by Oh et al [1] which you brought to our attention. It can be seen that KRNO achieves top performance on this benchmark on all four cases. Scripts for reproducing the results for the additional four benchmarks are provided in the anonymous repository under directories 'code/scripts/mujoco' and 'code/scripts/irregular_time_series/krno'.
>
> The results for MIMIC, USHCN, and Human Activity can be found in our response to reviewer **ntzg**.
>
> |Methods|Regular|30% dropped|50% dropped|70% dropped|
> |:---|:---:|:---:|:---:|:---:|
> |GRU-Δt|0.223 ± 0.020|0.198 ± 0.036|0.193 ± 0.015|0.196 ± 0.028|
> |GRU-D|0.578 ± 0.042|0.608 ± 0.032|0.587 ± 0.039|0.579 ± 0.052|
> |GRU-ODE|0.856 ± 0.016|0.857 ± 0.015|0.852 ± 0.015|0.861 ± 0.015|
> |ODE-RNN|0.328 ± 0.225|0.274 ± 0.213|0.237 ± 0.110|0.267 ± 0.217|
> |Latent-ODE|0.029 ± 0.011|0.056 ± 0.001|0.055 ± 0.004|0.058 ± 0.003|
> |Augmented-ODE|0.055 ± 0.004|0.056 ± 0.004|0.057 ± 0.005|0.057 ± 0.005|
> |ACE-NODE|0.039 ± 0.003|0.053 ± 0.007|0.053 ± 0.005|0.052 ± 0.006|
> |NCDE|0.028 ± 0.002|0.027 ± 0.000|0.027 ± 0.001|0.026 ± 0.001|
> |ANCDE|0.026 ± 0.001|0.025 ± 0.001|0.025 ± 0.001|0.024 ± 0.001|
> |EXIT|0.026 ± 0.000|0.025 ± 0.004|0.026 ± 0.000|0.026 ± 0.001|
> |LEAP|0.022 ± 0.002|0.022 ± 0.001|0.022 ± 0.002|0.022 ± 0.001|
> |Neural SDE|0.028 ± 0.004|0.029 ± 0.001|0.029 ± 0.001|0.027 ± 0.000|
> |Neural LSDE|0.013 ± 0.000|0.014 ± 0.001|0.014 ± 0.000|*0.013 ± 0.001*|
> |Neural LNSDE|*0.012 ± 0.001*|0.014 ± 0.001|0.014 ± 0.001|0.014 ± 0.000|
> |Neural GSDE|0.013 ± 0.001|*0.013 ± 0.001*|*0.013 ± 0.000*|0.014 ± 0.000|
> |KRNO|**0.007 ± 0.002**|**0.008 ± 0.002**|**0.011 ± 0.004**|**0.012 ± 0.002**|
>
>
> [1] Oh, Y., Lim, D., & Kim, S. Stable Neural Stochastic Differential Equations in Analyzing Irregular Time Series Data. ICLR 2024.
>
> ## 3. Questions and Weaknesses:
> **\[Q1\]** Thank you for your suggestion. We have restructured the methods section to highlight our contributions more clearly. We have also updated the related work section by discussing  SOTA methods such as Neural SDEs, T-PatchGNN, NeuralCDE, and Latent-ODE for irregularly sampled time-series forecasting.
>
> **\[Q2\].** Please refer to our response in section *Baselines and Evaluation Criteria* regarding comparison with SOTA methods and additional benchmarks on irregularly sampled time series datasets with missing observations. For these additional benchmarks, we have now added the results from multiple runs with different random seeds. It can be seen from the results that KRNO achieves new SOTA performance on three of the four new benchmarks we studied.
>
> **\[Q3\]** Neural operators have been primarily successful on data from physical systems governed by PDEs. Our comprehensive experiments demonstrate that KRNO generalizes effectively to both physical systems (MuJoCo) and non-physical datasets (healthcare, climate, human activity) with irregular sampling patterns.
>
> **\[W6\]** We have compared the memory usage and runtime analysis of KRNO with Neural GSDE for the MuJoCo dataset. For a batch size of 128, KRNO uses 850MB of memory while Neural GSDE uses 306MB of memory. On the GTX4090 GPU, for this setting, we found that the default KRNO configuration is around 10 times faster than Neural GSDE per iteration.

---

> > ### Comment · Reviewer_kNx6 · 2025-04-07
> >
> > I truly appreciate the authors' efforts in the rebuttal. In light of their responses and new results on real-world irregular datasets, as well as their initial experiments, I notice the performance improvements are mostly minor compared to baselines. Assessing the overall impact of the contribution (not substantially new theoretical results and experimental improvements), I prefer to maintain my initial scores.

---

> > > ### Author Response · Authors · 2025-04-08
> > >
> > > Thank you for your continued engagement with our work. Upon reflection, we wonder if there might have been a misinterpretation of our experimental results. The 46% error reduction on regular data and 38% on irregularly sampled data (MuJoCo) represent substantial improvements in forecasting accuracy - improvements that would typically be considered significant advances in the field. In our experience, improvements of even 5% over SOTA is often considered as a meaningful contribution.
> > >
> > > To clarify these substantial improvements:
> > >
> > > - On the MuJoCo benchmark, KRNO achieves a **46%** error reduction compared to Neural GSDE [1], the previous SOTA method published at **ICLR 2024** (Neural GSDE: **0.013** vs. KRNO: **0.007** MSE)
> > > - With 30% dropped observations, KRNO maintains a **38%** error reduction (**0.008** vs **0.013** MSE)
> > > - KRNO **consistently outperforms all 15** baseline methods across all dropout settings
> > > - Additionally, KRNO achieved SOTA performance on **3 of 4** irregular time-series benchmarks (MIMIC, USHCN, Human Activity, and MuJoCo), demonstrating its effectiveness across diverse domains with missing observations. On the MIMIC, USHCN, and Human Activity benchmarks, we compare against TimesNet, PatchTST, GRU-D, Warpformer, mTAND, Latent-ODE, and T-PatchGNN [2] - a recent study from **ICML 2024**.
> > >
> > > [1] Oh, Y., Lim, D., & Kim, S. "Stable Neural Stochastic Differential Equations in Analyzing Irregular Time Series Data." *ICLR 2024*.
> > >
> > > [2] Zhang et al., "Irregular multivariate time series forecasting: A transformable patching graph neural networks approach." *ICML 2024*.
> > >
> > > To ensure full *reproducibility* of these results, we have updated our anonymized repository with all code and scripts used for these irregularly sampled time-series benchmarks, including detailed instructions for replicating our experiments.
> > >
> > > Remarkably, we achieved these substantial improvements using the default KRNO architecture [3] without any dataset-specific hyperparameter tuning. This "out-of-the-box" performance stands in stark contrast to competing methods that typically require extensive tuning for each dataset. Such exceptional generalization across diverse data distributions suggests KRNO translates to meaningful practical improvements in forecasting accuracy, particularly for applications requiring precise predictions from irregularly sampled data, such as healthcare monitoring and climate science.
> > >
> > > Our work contributes not only empirical advances but also a novel operator-theoretic framework for handling irregularly sampled data, which opens new research directions. This framework enables continuous representations in both space and time without requiring specialized solvers or numerical integration schemes. We believe our numerical studies, spanning **24 diverse datasets** across multiple domains, demonstrate the broad applicability and effectiveness of our approach, particularly for the challenging irregular sampling settings that were the focus of your initial review.
> > >
> > > Given these objective metrics and the theoretical framework we have developed, we respectfully invite you to reconsider your assessment. In the spirit of scientific evaluation, where quantitative improvements of 46% over recent SOTA methods would typically be considered substantial contributions, we believe our work makes a meaningful advance to the field. We value your expertise and perspective on what would constitute a significant improvement in this domain.
> > >
> > > ###### [3] *The default KRNO architecture used in all experiments has three kernel integral layers with 20 channels each, lifting and projection layers are parametrized by MLPs with one hidden layer containing 128 hidden units, and the kernels in the integral layers are parametrized by MLPs with 3 hidden layers.*

---

### Official Review · Reviewer_dfXC · 2025-03-13

**Overall Recommendation:** 3

**Summary:**

The paper introduces a novel operator-theoretic approach for time-series forecasting by learning a continuous time-shift operator. This method provides a more flexible alternative to traditional autoregressive models, which rely on discrete time lags. The authors propose Khatri-Rao Neural Operators (KRNOs) as a new architecture to parametrize non-stationary integral transforms, enabling efficient learning of time-dependent dynamics in both temporal and spatio-temporal forecasting.

**Claims And Evidence:**

The paper presents several claims, most of which are backed by clear empirical results.

**Essential References Not Discussed:**

No

**Experimental Designs Or Analyses:**

The paper evaluates KRNO on several temporal and spatio-temporal datasets, including Darts, M4, and physics-based problems (Darcy flow, hyper-elastic problems). It compares KRNO with FNO, DeepONet, and other neural operators. It reports relative error as the primary evaluation metric. The paper claims that KRNO has almost linear computational complexity. However, there is no explicit runtime comparison with other baselines, making this claim difficult to verify.

**Methods And Evaluation Criteria:**

The paper evaluates KRNO on 29 different forecasting tasks, covering applications in climate modeling, financial markets, and fluid dynamics. The model is tested against leading baselines, including Fourier Neural Operators (FNO), DeepONet, and traditional autoregressive models

**Other Comments Or Suggestions:**

N\A

**Other Strengths And Weaknesses:**

The introduction of Khatri-Rao decompositions enhances computational efficiency compared to FNO and DeepONet, making it feasible for large-scale forecasting. The claim of near-linear complexity is promising, though a more explicit runtime analysis would strengthen this claim. The model is evaluated on 29 forecasting tasks, including climate modeling, physics simulations, showing broad applicability.

The claim that KRNO achieves near-linear complexity is reasonable based on Khatri-Rao properties, but the paper does not provide wall-clock runtime comparisons with FNO, DeepONet, and transformer-based methods and Scalability analysis for increasing dataset sizes and higher-dimensional problems.

**Questions For Authors:**

1. Can you provide runtime benchmarks comparing KRNO with FNO, DeepONet, and standard transformer models?
2. How does KRNO handle missing or noisy data?
3. Does KRNO generalize well to very high-dimensional spatio-temporal problems, such as 3D weather forecasting or turbulence modeling?

**Relation To Broader Scientific Literature:**

The continuous time-shift operator proposed in this paper generalizes discrete autoregressive models by modeling entire function trajectories instead of relying on discrete time steps. This idea extends prior work on neural operators like Fourier Neural Operators and DeepONet

**Theoretical Claims:**

The proofs for Proposition 2.1 and its generalization appear correct in principle and are logically structured, but they lack some explicit derivations and justifications for key claims (especially computational efficiency). While these omissions do not invalidate the results, addressing them would increase clarity.

---

> ### Author Rebuttal · Authors · 2025-04-01
>
> Thank you for your feedback and comments.
>
>
> ## 1. Computational Complexity and Runtime Analysis
>
> In Appendix G, we provide a detailed comparison of computational complexity and runtime between KRNO and FNO-3D using the spatio-temporal shallow water problem. While our initial analysis used the default 12 Fourier modes for FNO-3D across all spatial resolutions, we note that when increasing the number of modes for high-resolution datasets, FNO-3D's memory usage and training time exceed KRNO's requirements. The table below presents an updated runtime analysis comparing KRNO with FNO-3D across different spatial resolutions (S × S) using the maximum number of Fourier modes (S/2+1). These results demonstrate that KRNO is both faster and more memory-efficient than FNO-3D at higher resolutions.
>
> | Spatial resolution (S x S) | Memory (MB) | | | | Time (seconds) | | | |
> |:---:|:---:|:---:|:---:|:---:|:---:|:---:|:---:|:---:|
> | | Training | | Testing | | Training | | Testing | |
> | | **KRNO** | FNO-3D| **KRNO** | FNO-3D | **KRNO** | FNO-3D | **KRNO** | FNO-3D |
> | 32 × 32 | 1,390 | 1,074 | 708 | 1,074 | 0.0279 | 0.0234 | 0.0107 | 0.0062 |
> | 64 × 64 | 2,776 | 2,772 | 1,314 | 2,772 | 0.0394 | 0.0347 | 0.0134 | 0.0071 |
> | 96 × 96 | 4,884 | 5,264 | 2,366 | 5,264 | 0.0626 | 0.0710 | 0.0185 | 0.0149 |
> | 128 × 128 | 7,608 | 8,994 | 3,796 | 8,994 | 0.0999 | 0.1297 | 0.0312 | 0.0288 |
> | 160 × 160 | 10,040 | 13,764 | 5,644 | 11,288 | 0.1584 | 0.2088 | 0.0502 | 0.0456 |
>
> ## 2. Handling Missing or Noisy Data
>
> We now include additional experiments on challenging irregularly sampled time series benchmark datasets (MIMIC, USHCN, Human Activity, and MuJoCo), which contain missing and noisy observations. The performance of KRNO is compared against a range of alternative approaches with SOTA performance such as T-PatchGNN [1], NeuralSDE [2], NeuralCDE, PatchTST, and Latent-ODE.
>
> KRNO achieves new SOTA performance on three of the four benchmarks; please see the tables in the *Baselines and Evaluation Criteria* section of our responses to reviewers **ntzg** and **kNx6**. From these studies, we see that KRNO is able to handle missing and noisy data. In our future work, we will extend KRNO to support missing data for the case of missing data in spatio-temporal problems.
>
> [1] Zhang, Weijia, et al. "Irregular multivariate time series forecasting: A transformable patching graph neural networks approach." ICML 2024.
>
> [2] Oh, Y., Lim, D., & Kim, S. Stable Neural Stochastic Differential Equations in Analyzing Irregular Time Series Data. ICLR 2024.
>
> ## 3. Generalization to High-Dimensional Spatio-Temporal Problems
>
> Numerical studies on spatial modeling problems, 2D spatio-temporal forecasting problems, and temporal forecasting problems show that the KRNO architecture provides better performance than competing methods for this class of problems such as FNO, DeepONet, and LOCA. We plan to evaluate KRNO on challenging 3D spatio-temporal problems in future work.

---

### Official Review · Reviewer_ntzg · 2025-03-13

**Overall Recommendation:** 3

**Summary:**

The paper introduces a novel operator-theoretic framework for time-series forecasting, leveraging the Khatri-Rao Neural Operator (KRNO) to learn continuous time-shift operators. By relaxing the discrete lag factor in autoregressive models, KRNO enables super-resolution forecasting in both space and time while handling irregularly sampled observations. The authors demonstrate KRNO’s scalability and competitiveness across benchmark datasets.

**Claims And Evidence:**

1. Efficiency: KRNO achieves near-linear computational cost for non-stationary integral transforms, scaling better than FNO and other neural operators.

2. Flexibility: KRNO handles irregular sampling and super-resolution forecasting by parametrizing the time-shift operator as a continuous kernel.

3. Superior Performance: KRNO ranks among the top 3 methods on 21/29 test cases and tops 10/29 datasets.

Weakness:
1. Overstated Generality: The claim that KRNO “inherits the benefits of neural operators” (e.g., discretization independence) overlooks its reliance on specific kernel structures (Equation 7).

2. Lack of Theoretical Guarantees: While Proposition 2.1 supports computational efficiency, there is no theoretical analysis of approximation error or stability for non-product-kernel scenarios.

**Essential References Not Discussed:**

No

**Experimental Designs Or Analyses:**

1. Lack of Hyperparameter Tuning Details: Key parameters (e.g., number of layers, hidden units) are not fully documented, reducing reproducibility.

2. Inconsistent Evaluation Protocols: Some experiments (e.g., M4) use recursive forecasting, while others (e.g., Darts) rely on fixed window sizes, complicating comparisons.

3. Missing Baselines: Notable omissions include recent SOTA methods like TimesNet (Wu et al., 2022) and LogTrans (Li et al., 2023) for time-series forecasting.

4. Computational Cost Analysis: While Table 7 compares GPU memory, training/inference times are only shown for shallow water (Figure 9), lacking scalability analysis for larger datasets (e.g., M4).

**Methods And Evaluation Criteria:**

1. KRNO is built on non-stationary integral transforms with component-wise kernels decomposed as Khatri-Rao products (Equation 7).

2. The architecture includes lifting/projection layers and three kernel integral transform layers, with neural networks parametrizing each kernel.

Weakness:
1. Assumption of Product Grids: KRNO’s near-linear complexity relies on input/output data lying on product grids (e.g., time × latitude × longitude). This limits applicability to irregularly gridded or high-dimensional data.

2. Fixed Hyperparameters: The time-shift operator’s boundaries (t_p, t_f) are treated as hyperparameters, but the paper does not discuss adaptive strategies for varying forecasting horizons.

3. Ignoring Temporal Dependencies: The operator learns mappings over fixed windows ([t_p, t] to (t, t_f]), neglecting long-range dependencies beyond the window size.

**Other Comments Or Suggestions:**

1. Clarify the scope of KRNO’s applicability (e.g., product grids vs. arbitrary grids).

2. Add error bounds or stability analysis for non-stationary kernels.

3. Include recent SOTA methods (e.g., TimesNet, LogTrans) in benchmarks.

4. Provide full hyperparameter details and reproducibility protocols.

**Other Strengths And Weaknesses:**

See review above

**Questions For Authors:**

See review above

**Relation To Broader Scientific Literature:**

Connects KRNO to operator-learning frameworks (DeepONet, FNO) and highlights advantages over autoregressive models (e.g., Transformer, N-BEATS).

**Theoretical Claims:**

There is no theoretical analysis of how approximation errors in the kernel decomposition (Equation 7) propagate to the overall forecasting error.
While KRNO avoids ODE-based adjoints, the paper does not compare its gradient estimation stability to neural ODEs in noisy or high-dimensional settings.

---

> ### Author Rebuttal · Authors · 2025-04-01
>
> Thank you for your feedback and comments.
>
> ## 1. Theoretical Guarantees and Kernel Structure
>
> The computational complexity of the kernel integral transform layer scales as O($n^2$), where $n$ is the number of quadrature nodes in the input and output domains. Additional assumptions are required to reduce this complexity. For instance, FNOs assume that the kernel is stationary, thereby enabling the use of FFT. In the present work, we overcome this complexity using product structured non-stationary kernels.
>
> The only approximation error comes from the quadrature scheme used to compute the integral in equation 6; the non-stationary kernel is evaluated exactly, without additional errors. We have added an ablation study to show the influence of the number of quadrature points on the generalization performance of KRNO (please refer to our response to reviewer **P25q** on ablation studies). A particularly attractive feature of KRNO is that it allows us to learn a flexible non-stationary kernel for each component of the product structured kernel. As evidenced by our numerical studies, this additional flexibility provides performance gains over stationary kernel-based neural operator architectures, while significantly reducing the model parameter count.
>
>
> ## 2. Implementation and Applicability
>
> In our exposition of KRNO, we considered the case of product grids to achieve almost linear scalability. It is worth noting that our approach can be extended to unstructured grids by introducing a learnable function that maps the data from an unstructured grid to a latent product grid as in Geo-FNO [1].
>
> We have added the missing hyper-parameter details for each benchmark in the Appendix. To ensure reproducibility, we have shared the source code and scripts used in our numerical studies at [https://anonymous.4open.science/r/KRNO-1F4F/](https://anonymous.4open.science/r/KRNO-1F4F/).
>
> [1] Li, Zongyi, et al. "Fourier neural operator with learned deformations for pdes on general geometries." JMLR 24.388 (2023): 1-26.
>
>
> ## 3. Baselines and Evaluation Criteria:
>
> We agree that numerical studies on regularly sampled time-series do not allow a clear demonstration of the full capabilities offered by KRNO. We now include additional experiments on challenging irregularly sampled time series benchmark datasets (MIMIC, USHCN, Human Activity, and MuJoCo) and the results are compared against a range of alternative approaches with SOTA performance such as T-PatchGNN [1], NeuralSDE [2], NeuralCDE, PatchTST, and Latent-ODE. KRNO achieves new SOTA performance on three of the four benchmarks; please see the Table below which provides results for three of the datasets. The results for the MuJoCo benchmark are provided in our response to reviewer **kNx6**.
>
> | **Method** | **MIMIC MSE×10⁻²** | **MIMIC MAE×10⁻²** | **USHCN MSE×10⁻¹** | **USHCN MAE×10⁻¹** | **Human Activity MSE×10⁻³** | **Human Activity MAE×10⁻²** |
> |:---:|:---:|:---:|:---:|:---:|:---:|:---:|
> | TimesNet | 5.88 ± 0.08 | 13.62 ± 0.07 | 5.58 ± 0.05 | 3.60 ± 0.04 | 3.12 ± 0.01 | 3.56 ± 0.02 |
> | PatchTST | 3.78 ± 0.03 | 12.43 ±0.10 | 5.75 ± 0.01 | 3.57 ± 0.02 | 4.29 ± 0.14 | 4.80 ± 0.09 |
> | GRU-D | 1.76 ± 0.03 | 7.53 ± 0.09 | 5.54 ± 0.38 | 3.40 ± 0.28 | 2.94 ± 0.05 | 3.51 ± 0.06 |
> | Warpformer | 1.73 ± 0.04 | 7.58 ± 0.13 | 5.25 ± 0.05 | 3.23 ± 0.05 | *2.79 ± 0.04* | *3.39 ± 0.03* |
> | mTAND | 1.85 ± 0.06 | 7.73 ± 0.13 | 5.33 ± 0.05 | 3.26 ± 0.10 | 3.22 ± 0.07 | 3.81 ± 0.07 |
> | Latent-ODE | 1.89 ± 0.19 | 8.11 ± 0.52 | 5.62 ± 0.03 | 3.60 ± 0.12 | 3.34 ± 0.11 | 3.94 ± 0.12 |
> | T-PatchGNN | *1.69 ± 0.03* | **7.22 ±0.09** | *5.00 ± 0.04* | *3.08 ± 0.04* | **2.66 ± 0.03** | **3.15 ± 0.02** |
> | KRNO | **1.57 ± 0.02** | *7.43 ± 0.06* | **4.95 ± 0.08** | **3.06 ± 0.08** | 2.85 ± 0.03 | 3.46 ± 0.02 |
>
> [1] Zhang, Weijia, et al. "Irregular multivariate time series forecasting: A transformable patching graph neural networks approach." ICML 2024.
>
> [2] Oh, Y., Lim, D., & Kim, S. Stable Neural Stochastic Differential Equations in Analyzing Irregular Time Series Data. ICLR 2024.
>
>
> ## 4. Computational Cost Analysis
>
> Please refer to our first response to reviewer **dfXC** for more clarification on the computational complexity and runtime analysis of KRNO compared to FNO.
>
> ## 5. Temporal Dependency
>
> You have raised a very important point about the choice of $t_p$ and $t_f$. In our numerical studies, we treated them as hyperparameters. However, it would be valuable to explore how these parameters can be adaptively chosen and how this could potentially improve generalization. This point is also closely related to the application of KRNO to model long-range dependencies. We plan to pursue these directions in future work.

---

### Official Review · Reviewer_P25q · 2025-03-20

**Overall Recommendation:** 3

**Summary:**

This paper presents a method for time-series forecasting that treats the task as learning a continuous time-shift operator, approximated via a proposed architecture called Khatri-Rao Neural Operator (KRNO). The operator is modeled as an integral transform with a non-stationary kernel, decomposed via Khatri-Rao product structure. The resulting model enables forecasting with irregularly sampled data, super-resolution, and low parameter count. The method is validated across a large suite of temporal and spatio-temporal benchmarks.

**Claims And Evidence:**

The main claim of the authors is that they introduce the continuous time-shift operator. There is a drastic disconnect between this claim and the empirical studies. None of the studies demonstrate the importance of continuous time-shift as concept. The continuos shift in either time or space can be achieved by a simple MLP, mapping time and coordinates along with some additional features into predictions. Why do we need KRNO, what is special about it that other architectures cannot do? This is further emphasized by the fact that large part of experiments are conducted on regularly sampled non-spatial datasets that have so many methods that work extremely well on them, that the authors will not have enough space in a 20-page paper to review, explain and present their results. May I suggest that the authors rather focus on a very concrete problem and drill it down, instead of overwhelming the reader with results and studies that are largely irrelevant from the point of view of the problem that they claim to solve?

**Essential References Not Discussed:**

[1] Horn et al., Set Functions for Time Series https://arxiv.org/pdf/1909.12064
Offers a non-sequential way to model irregular time series using permutation-invariant functions.

[2] Kidger et al., Neural Controlled Differential Equations for Irregular Time Series, https://proceedings.neurips.cc/paper/2020/file/4a5876b450b45371f6cfe5047ac8cd45-Paper.pdf,
memory-efficient ODE-based approach to irregular time-series

**Experimental Designs Or Analyses:**

- The description of many experiments lacks clarity. For example, for Darcy-flow and hyper-elastic benchmark no details are provided at all. What are the problems, why are they relevant for the evaluation of the proposed algorithm? What are dataset/problem sizes, what are the splits? How is the L2-relative error computed, exactly? Similar remarks apply to most benchmarks used in the paper.

- The relevance of Figure 2 is not clear. It would constitute a much stronger case if it contained a comparison with baseline methods that significantly underperform on the presented anecdotes. The fact that the proposed technique apparently does reasonably well on these cases does not render convincing evidence of the proposed framework being able to solve problems that other approaches fail to provide adequate solutions for.

- The paper contains no ablation studies whatsoever. This is a red flag. What are important model components, key assumptions and their effects on model performance?

- Missing comparisons to neural controlled differential equations (CDEs), which are designed for irregular and continuous-time forecasting, are only discussed briefly and not benchmarked empirically.

**Methods And Evaluation Criteria:**

- I am not quite convinced that the use of regularly sampled forecasting datasets, such as Darts, Crypto, Baseball or M4, is relevant, as the proposed theory and operator-based approach seem to be most applicable to irregularly spaced and spatio-temporal data. In my view, these results overload the paper and distract attention from other important topics. For example, there is very little discussion of what the proposed model learns i.e., how the time-shift operator behaves or generalizes across inputs. Similarly, there is very little discussion in the text of the implementation details. How does the inference path of the model look like, exactly? Can you provide equations describing it? Additionally, there are no ablation studies.

**Other Comments Or Suggestions:**

NA

**Other Strengths And Weaknesses:**

NA

**Questions For Authors:**

- Can you clarify the distinction between kernels being functions of time/space vs. functions of the data? Does this violate the assumptions made in the integral operator formulation?

- Why are neural CDEs or latent ODEs not included in the empirical comparisons, especially given the shared goal of handling irregular sampling?

- Can you add ablation studies showing the effects of important model components on model performance? As an example, could you provide an ablation of the quadrature rule used (midpoint vs. trapezoidal vs. learned)? What are other important components and their effects on the model accuracy/speed/memory?

- Is the method capable of forecasting in the presence of measurement noise or exogenous inputs?

**Relation To Broader Scientific Literature:**

- Overuse of standard mathematical machinery: Much of the theoretical exposition leans heavily on classical results (e.g., Grönwall’s inequality, semigroup continuity), which could have been cited more concisely. Similarly, Proposition 2.1 is a simple consequence of the Khatri–Rao product structure and it does not need a formal statement in the form of proposition or a proof. Its inclusion, while technically correct, adds bulk without substantive new theory.

**Theoretical Claims:**

- Apparent theory/practice disconnect: the transition between theory (quadrature approximations, kernel formulation) and implementation (data-driven neural parametrization) feels abrupt (i.e. Line 213). The theory defines kernels as functions of coordinates, but the practical implementation uses learnable functions of data, raising questions about the operator-theoretic validity of the implementation.

---

> ### Author Rebuttal · Authors · 2025-04-01
>
> Thank you for your feedback and comments.
>
> ### 1. Evaluation on Irregularly Sampled Datasets
> We agree that numerical studies on regularly sampled time-series do not allow a clear demonstration of the full capabilities offered by KRNO. We now include additional experiments on challenging irregularly sampled time series benchmark datasets (MIMIC, USHCN, Human Activity, and MuJoCo) and the results are compared against a range of alternative approaches such as T-PatchGNN, NeuralSDE, NeuralCDE, PatchTST, and Latent-ODE. KRNO achieves SOTA performance on three of the four benchmarks. Please see Tables in section *Baselines and Evaluation Criteria* of our response to reviewers **ntzg** and **kNx6**.
> As suggested, we have updated the Appendix to include more details on the Darcy-flow and hyper-elasticity benchmarks.
>
> ### 2.  Model Architecture & Implementation Details
> KRNO's architecture follows the general structure of neural operators (lines 129-150), i.e., lifting layers followed by series of kernel integral layers and a projection layer. The key difference is our use of a non-stationary product structured kernel in the kernel integral transform (Equation 6). The KRNO representation of the continuous time-shift operator can be written as:
> $$
> \mathcal{A}_{t_p}^{t,t_f}:= \mathcal{P} \circ \mathcal{K}_n \circ \ldots \circ \mathcal{K}_1 \circ \mathcal{L},
> $$
> where $\mathcal{K}_i: \mathbb{R}^{p} \to \mathbb{R}^{q}$ is a kernel integral layer defined in Equation 5, $\mathcal{L}: \mathbb{R}^{n} \to \mathbb{R}^{c}$, and $\mathcal{P}: \mathbb{R}^{c} \to \mathbb{R}^{n}$ are the pointwise lifting and projection layers parameterized by neural networks with one hidden layer, and $c$ is the number of channels in integral layers.
>
> To ensure reproducibility, we have shared the source code and scripts used in our studies at [https://anonymous.4open.science/r/KRNO-1F4F/](https://anonymous.4open.science/r/KRNO-1F4F/).
>
> ### 3. Theoretical Consistency
> Please note that the matrix-valued kernels in KRNO are consistently defined as functions of time and space throughout the paper and in our implementation. The kernel integral layers in Equation 5 use these parametrized space-time kernels to learn mappings between function spaces. As a consequence, the numerical implementation is aligned with the operator-theoretic formulation.
>
> ### 4. Ablation Studies
> Thank you for suggesting insightful ablation studies. We have conducted ablation studies as suggested to better understand the key components that influence generalization. One of the key parameters is the number of integral layers/channels. Larger values of this parameter enhance model capacity at the expense of increased memory usage for high-resolution data. This trade-off can be effectively managed by adjusting the number of quadrature points in the kernel integral layers. The following ablation study on the Elasticity problem (learning stress fields from void deformation in elastic blocks) confirms this flexibility. Using training data given on a 41×41 grid, we notice that reducing latent grid resolution in the integral layers significantly decreases memory requirements with minimal impact on accuracy. We intend to include additional ablation studies to illustrate the impact of the quadrature rule on generalization performance.
> |Latent Grid Resolution|L2 Rel. error|GPU Memory (MB)|
> |:---:|:---:|:---:|
> |16 x 16 |5.20 ± 0.18 % |1,386 |
> |24 x 24 |5.14 ± 0.17 % |1,774 |
> |32 x 32 |5.14 ± 0.26 % |2,186 |
> |40 x 40 |5.12 ± 0.15 % |2,650 |
> |48 x 48 |5.16 ± 0.16 % |3,174 |
>
> ### 5. Handling Measurement Noise and Exogenous Inputs
> The new irregularly sampled benchmarks mentioned earlier are challenging with missing and noisy observations. Our results show that KRNO effectively handles such data. In the present work, we formulate the time-shift operator in the setting of deterministic ordinary/partial differential equations. This formalism enabled sufficient flexibility to provide strong generalization across temporal and spatio-temporal forecasting problems from diverse domains.
>
> For example, for the case of time-series datasets, the forced time-shift operator can be defined as $X\_{(t,t_f]}=\mathcal{A}\_{t_p}^{t,t_f} (X_{[t_p,t]}, f_{[t_p,t_f]})$, where $X_{[t_p,t]}$ denotes the state trajectory over the time-interval $[t_p,t]$ while $f_{[t_p,t_f]}$ denotes the forcing function over the input and forecasting time-interval $[t_p,t_f]$. To ensure that the time-shift operator is causal (i.e., the predictions made at time instant $\tau$ are not influenced by the future values of the forcing function), the integration domain of the kernel integral transform w.r.t. $f$ should be set to $[t_p, \tau]$, where $\tau \in (t, t_f]$. A similar approach can be used to define the time-shift operator for forced spatio-temporal dynamical systems. These extensions would enable our approach to be applied to systems with exogenous effects.

---

> > ### Comment · Reviewer_P25q · 2025-04-04
> >
> > I would like to thank the authors for providing a comprehensive response. I raise the score slightly based on the current improvements. I would still like the authors to provide more detail regarding the following questions.
> >
> > 1. Theoretical Consistency response is rather shallow. Can you please provide a draft of how the transition in the paper will be made from the kernel theory to the data-driven neural parametrization? Yes, kernels are functions of time and space. But they are not functions of data, aren't they? If you could justify this transition, preferably in a theoretically rigorous way (can you make a case with a theorem, for example?), this would significantly strengthen the paper.
> >
> > 2. The reliance of testing protocols on regularly sampled data brings a lot of bulk in the paper, but does it really demonstrate the key features of the proposed method? May I suggest that a bulk of these studies be moved to appendices? Can this leave more space for things such as the quadrature grid ablation? Finally, I do not believe this point has been addressed:
> > > For example, there is very little discussion of what the proposed model learns i.e., how the time-shift operator behaves or generalizes across inputs.
> > >
> > Can you use the space in the paper to actually show something unique about the method working with irregularly spaced data? My honest opinion is that it is very unlikely that the operator based approach designed to handle irregularly spaced spatio-temporal data will be used as a handy replacement for PatchTST or DLinear, or whatever other method optimized for dense regularly spaced data. Why even spend precious paper space emphasizing aspects that are not core to the contribution?
> >
> > 3. What is the forward path of the method, in terms of neural layers? Yes, the code helps to solve the reproducibility issue and I appreciate it. But I also want to understand how the operators translate into actual neural layers (i.e. matrix/bias/activation function operations applied to input tensor) and signal flows through the architecture. Can you present forward pass of the model in the paper using some language similar to this one: https://arxiv.org/pdf/1706.03762 as an example?
> >
> > 4. Much of the theoretical exposition leans heavily on classical results (e.g., Grönwall’s inequality, semigroup continuity), which could have been cited more concisely. Similarly, Proposition 2.1 is a simple consequence of the Khatri–Rao product structure and it does not need a formal statement in the form of proposition or a proof. Its inclusion, while technically correct, adds bulk without substantive new theory.

---

> > > ### Author Response · Authors · 2025-04-08
> > >
> > > We appreciate your thoughtful feedback and the opportunity to clarify aspects of our work.
> > >
> > > ## Theoretical Consistency
> > > The transition from theory to data-driven parametrization follows established principles in deep kernel methods, where neural networks learn mappings to a feature space where standard kernels are applied, with all parameters jointly learned.
> > >
> > > In KRNO, our kernels strictly maintain the form $k(x,t)$ where $x,t$ are space-time coordinates. To ensure expressivity, the kernel is parameterized by neural networks with learned weights & biases, i.e., $k_\theta(x,t)$, where $\theta = \text{argmin}\_\theta \ell(f(k_\theta),D)$ with $\ell$ denoting the loss function, $f_{k_\theta}$ the operator with kernel $k_\theta$, and $D$ the training data.
> > >
> > > As an illustrative example, consider a time-series $u(t) \in \mathbb{R}^n$ over $[t_p,t_f]$ modeled using a *single-layer KRNO* with no lifting and projection layers. Let $U_p = [u(t_1),u(t_2),u(t_3)]^T \in \mathbb{R}^{3 \times n}$ and $U_f = [u(t_4),u(t_5)]^T \in \mathbb{R}^{2 \times n}$ denote observations over the time-intervals $[t_p,t]$ (input) and $(t,t_f]$ (output), respectively. The KRNO prediction at time $t_j \in (t,t_f]$ takes the form
> > > $$
> > > \begin{align*}
> > > \hat{u}(t_j)=\int_{t_p}^{t} k_\theta (t_j,t')u(t')dt' &\approx \sum_{i=1}^{3}k_\theta (t_j,t_i)w_iu(t_i)=[k_\theta (t_j,t_1),k_\theta (t_j,t_2),k_\theta (t_j,t_3)]
> > > \begin{bmatrix}w_1u(t_1)\\\\w_2u(t_2)\\\\w_3u(t_3)\end{bmatrix},
> > > \end{align*}
> > > $$
> > > where $k_\theta: \mathbb{R} \times \mathbb{R} \to \mathbb{R}^{n \times n}$ is a matrix-valued kernel parametrized using a neural network and $w_i$ are quadrature weights.
> > > The prediction over $(t,t_f]$, i.e., $\hat{U}\_f =[\hat{u}(t_4),\hat{u}(t_5)]^T\in \mathbb{R}^{2\times n}$ becomes
> > > $$
> > > \begin{align*}
> > > \hat{U}\_f= \mathcal{K}(U_p)=k_\theta (T_f,T_p) \text{vec(diag}(w) U_p) =\begin{bmatrix}k_\theta (t_4,t_1) & k_\theta (t_4,t_2) &  k_\theta (t_4,t_3)\\\\ k_\theta (t_5,t_1) & k_\theta (t_5,t_2) & k_\theta(t_5,t_3)\end{bmatrix}
> > > \begin{bmatrix} w_1u(t_1)\\\\ w_2u(t_2)\\\\ w_3u(t_3)\end{bmatrix}
> > > \end{align*},
> > > $$
> > > where $k_\theta(T_f, T_p) \in \mathbb{R}^{2n \times 3n}$ is the kernel matrix evaluated at the quadrature nodes $T_f=[t_4,t_5]$ and $T_p=[t_1,t_2,t_3]$.
> > >
> > > The parameters $\theta$ are learned by minimizing the $L^2$ error between $\hat{U}_f$ and the observed $U_f$.
> > >
> > > Our kernel parametrization approach preserves the operator-theoretic formulation (kernel remains a function of coordinates) while enabling data-driven adaptation through the learned parameters $\theta$.
> > >
> > > ## Reorganizing benchmarks and revisions
> > > Thank you for your suggestion - we will restructure the paper to focus on irregularly sampled data in the main text.
> > >
> > > In the revised paper, we will emphasize KRNO's uniqueness stemming from its continuous representation in both space and time, offering several key advantages, particularly for irregularly sampled data. For example, at each prediction time $t_j$, the kernel $k_\theta(t_j, t')$ *learns to identify and weight the most relevant historical time points*, without requiring regular sampling. Moreover, the multi-layer structure allows KRNO to discover both local temporal dynamics and global patterns (through the composition of multiple kernel layers). Figure 1 demonstrates this capability where we show super-resolution in both space and time.
> > >
> > > ## Forward pass of KRNO
> > > We like your suggestion of graphically illustrating the KRNO forward pass, identifying tensor shapes. We will include this in the revised paper. To illustrate the steps compactly, consider a *single* KRNO layer which includes a pointwise lifting and projection layer with $n$ channels applied to the same example described earlier. Given input sequence $U_p\in \mathbb{R}^{3\times n}$, KRNO predicts $\hat{U}_f=[\hat{u}(t_4),\hat{u}(t_5)]^T\in \mathbb{R}^{2\times n}$ as:
> > >
> > > $$\hat{U}_f=\mathcal{P}\circ\mathcal{K}\circ\mathcal{L}(U_p).$$
> > >
> > > Parametrizing the lifting and projection layers using an MLP with one hidden layer (for brevity), the matrix operations in the forward pass are:
> > >
> > > - Lifting layer: $U_1= \mathcal{L}(U_p)=(\sigma(U_p W_{l_1}+b_{l_1}))W_{l_2}+b_{l_2}$, where $U_1 \in \mathbb{R}^{3\times n}$
> > >
> > > - Integral layer: $U_2'=\mathcal{K}(U_1)= k_\theta(T_f, T_p) \text{vec(diag}(w) U_1), $ where $k_\theta(T_f, T_p) \in \mathbb{R}^{2n\times 3n}$ and $U_2' \in \mathbb{R}^{2n}$
> > >
> > > - $U_2=\text{Reshape}(U_2') \in \mathbb{R}^{2\times n}$
> > >
> > > - Projection layer: $\hat{U}\_f=\mathcal{P}(U_2)=(\sigma(U_2 W_{p_1} + b_{p_1})) W_{p_2}+b_{p_2}$, where $\hat{U}_f \in \mathbb{R}^{2\times n}$
> > >
> > > Here, $W_{l_1},W_{l_2},W_{p_1},W_{p_2}\in\mathbb{R}^{n\times n}$ and $b_{l_1},b_{l_2},b_{p_1},b_{p_2}\in\mathbb{R}^{n}$ are weights and biases.
> > >
> > > ## Streamlining theoretical exposition.
> > > We appreciate your suggestions about the theoretical presentation. In the revised manuscript, we will focus on aspects unique to our contribution.
> > >
> > > Thank you again for your insightful feedback that will help us improve the manuscript.

---

### Decision · Program_Chairs · 2025-05-01

**Decision:**

Accept (poster)

**Comment:**

The paper proposes a continuous in-time learning for operators called Khatri-Rao Neural Operator (KRNO). The operator is modeled as an integral transform with a non-stationary kernel, which has also been studied in the past but, this time, decomposed via the Khatri-Rao product structure. The resulting model enables forecasting with irregularly sampled data and super-resolution.

The authors are encouraged to clarify in their papers that there are earlier works that considered mapping from time interval to time interval in function spaces, making the underlying kernels to be time dependent. The early studies of these sorts are UNO and "Tipping Point Forecasting in Non-Stationary Dynamics on Function Spaces".